# An Update on Techniques to Assess Normal Mode Behavior of Rock Arches by Ambient Vibrations

Mauro Häusler[1], Paul R. Geimer[2], Riley Finnegan[2], Donat Fäh[1], Jeffrey R. Moore[2]

[1]Swiss Seismological Service, ETH Zurich, Zurich, 8092, Switzerland

[2]Department of Geology and Geophysics, University of Utah, Salt Lake City, 84112, USA

*Correspondence to*: Mauro Häusler (mauro.haeusler@sed.ethz.ch)

## Abstract

Natural rock arches are rare and beautiful geologic landforms with important cultural value. As such, their management requires periodic assessment of structural integrity to understand environmental and anthropogenic influences on

arch stability. Measurements of passive seismic vibrations represent a rapid and non-invasive technique to describe the dynamic properties of natural arches, including resonant frequencies, modal damping ratios, and mode shapes, which can be monitored over time for structural health assessment. However, commonly applied spectral analysis tools are often limited in their ability to resolve characteristics of closely spaced or complex higher-order modes. Therefore, we investigate two techniques well-established in the field of civil engineering through application to a set of natural arches previously

characterized using polarization analysis and spectral peak-picking techniques. Results from Enhanced Frequency Domain Decomposition and parametric Covariance-driven Stochastic Subspace Identification modal analyses showed generally good agreement with spectral peak-picking and frequency-dependent polarization analyses. However, we show that these advanced techniques offer the capability to resolve closely spaced modes including their corresponding modal damping ratios. In addition, due to preservation of phase information, enhanced frequency domain decomposition allows for direct and convenient

three-dimensional visualization of mode shapes. These techniques provide detailed characterization of dynamic parameters, which can be monitored to detect structural changes indicating damage and failure, and in addition have the potential to improve numerical models used for arch stability assessment. Results of our study encourage broad adoption and application of these advanced modal analysis techniques for dynamic analysis of a wide range of geological features.

## 1 Introduction

Natural rock arches form by erosion (Bruthans et al., 2014;Ostanin et al., 2017) and are major tourist attractions worldwide. However, ongoing weathering can lead to partial or complete collapse posing a hazard to visitors; prominent examples include collapse of London Bridge (Australia) in 1990 (Woodroffe, 2002), rockfall from Landscape Arch (USA) in 1991 and 1995 above a hiking trail (Deseret News, 1991), and collapse of the Azure Window in Malta in 2017 (Satariano and Gauci, 2019). As arches occur in a variety of forms and settings, simple tools for stability assessment do not exist, and current

practices often include site-specific geomechanical characterization and numerical modeling (Budetta et al., 2019). In recent decades, the stability of engineered structures, such as buildings and bridges, has been increasingly analyzed using measurements of their vibrational properties associated with resonance. Understanding this dynamic response to ambient loading forms the basis for the field of structural health monitoring (SHM, Doebling et al., 1996). More recently, SHM concepts have been applied at natural rock arches and other geological formations to improve site characterization and hazard assessment

associated with failure of these features (e.g., Bottelin et al., 2013;Burjánek et al., 2018;Iannucci et al., 2020;Kleinbrod et al., 2019;Mercerat et al., 2021;Moore et al., 2018). Passive seismic measurements then provide a non-invasive means to monitor dynamic behavior and evaluate stability in the presence of natural or anthropogenic stimuli, which is especially valuable at culturally important sites where more invasive or destructive monitoring techniques (e.g., taking rock samples) may not be permitted.

Passive stability assessment often involves repeated or continuous measurements of a structure to monitor deviations in baseline structural dynamic behavior. This dynamic behavior is characterized by natural frequencies, corresponding mode shapes (i.e., structural deflection at those frequencies) and damping ratios (e.g., Chopra, 2015). While damping describes internal energy dissipation and radiation out of the system, resonant frequencies are primarily a function of stiffness and mass. As the mass of a rock landform is approximately constant over time (in the absence of mass wasting events), variations in

resonant frequencies arise primarily due to changes in rock mass stiffness, which can be associated with rock damage and environmental influences, such as temperature and moisture (Colombero et al., 2021;Bottelin et al., 2013). As internal crack growth accumulates during progressive failure, stiffness and thus frequencies are anticipated to decrease; for example, Lévy et al. (2010) reported a drop in resonant frequency of about 20% less than two weeks prior to collapse of a 21,000 m³ rock column, which they attributed to progressive failure. More quantitative assessments of stress conditions prior to failure require

individual features to be numerically modeled with realistic values for rock density and Young's modulus. With density constrained by rock samples, Young's modulus can be derived from dynamic properties by minimizing the error between observed and modeled resonance attributes (Moore et al., 2018;Moore et al., 2020;Geimer et al., 2020). Such model validation facilitates the estimation of the three-dimensional stress field, used by Moore et al. (2020) to identify arches that may be more prone to tensile crack growth and structural failure.

Modal analysis techniques used in structural health monitoring of geological features rely primarily on identification of spectral attributes from in-situ ambient vibration data. Power spectra visualizations provide a means for first interpretation, often leading to identification of resonant frequencies that can be confirmed through numerical modeling (Moore et al., 2018), while site-to-reference spectral ratios may be used to eliminate source and path effects in order to identify and track resonant frequencies (e.g., Weber et al., 2018). Selecting the maximum peak directly on the power spectra to determine the resonant

frequency is usually referred to as "peak-picking". By additionally selecting the frequencies left and right of the resonant peak, where the power drops by 3dB, a simple estimate of the modal damping ratio can be obtained (see Section 3.1 for details).

    Frequency-dependent polarization analysis (PA) provides a tool to estimate the modal deflection at resonance (Burjánek et al., 2012;Geimer et al., 2020). However, these spectral analysis techniques fall short when applied to more

complex systems, such as cases with closely spaced and overlapping modes, which have identical or similar frequencies but different mode shapes. In addition, phase information is not preserved across separate recording stations, impeding precise determination of mode shapes for higher modes. Thus, new techniques are necessary for refined modal analysis supporting structural health monitoring of rock landforms and providing accurate input parameters for stability assessment using numerical models. Among these, Enhanced Frequency Domain Decomposition (EFDD, Brincker et al., 2001a;Brincker et al., 2001b) is a promising approach to identify resonant frequencies, damping, and polarization attributes, and is well-suited to distinguish closely spaced modes. The Covariance-driven Stochastic Subspace Identification (SSI-COV) is an alternative time-domain technique that is especially beneficial for accurate estimates of modal damping ratios (van Overschee, 1996). Since their introduction, both techniques have become standard methods for analysis of engineered structures (e.g., Brincker and Ventura, 2015), and have been compared, yielding similar results (Cheynet et al., 2017;Brincker et al., 2000). Using these complementary techniques, Bayraktar et al. (2015) found a good agreement between EFDD and SSI-COV in their study on historical masonry arch bridges with resonant frequencies and damping ratios comparable to the natural rock arches and bridges studied here. Furthermore, frequency domain decomposition has been applied on natural features, such as sedimentary valleys, glaciers, and rock slope instabilities (Poggi et al., 2015;Preiswerk et al., 2019;Häusler et al., 2021, 2019;Ermert et al., 2014), while application of SSI-COV has remained restricted to engineered structures.

In this study, we analyze the modal characteristics of four natural rock arches in Utah (USA), previously investigated by Geimer et al. (2020). As these arches exhibit various spectral complexities which complicate dynamic analyses, we apply two operational modal analysis techniques – EFDD and SSI-COV – to improve identification and characterization of normal modes. Our results highlight the value and versatility of EFDD and SSI-COV for structural characterization and monitoring in geologic hazard applications, which we propose is useful across a broad range of geomorphologic features beyond our studied landforms, such as rock slope instabilities and rock towers (Bottelin et al., 2013;Häusler et al., 2021;Moore et al., 2019).

## 2 Data acquisition and study sites

Ambient vibration data processed in this study were collected at four natural rock arches in Utah by Geimer et al. (2020). These consist of three single-station measurements conducted using a Nanometrics Trillium Compact 20-s seismometer (TC 20-s, sites: Rainbow Bridge, Corona Arch, Squint Arch) and two array measurements using three-component Fairfield Zland 5-Hz nodal geophones with synchronous recording (sites: Squint Arch and Musselman Arch). Table 1 summarizes the arch measurements, including data acquisition length, site coordinates, and number of sensors deployed. Prior to processing, all data were corrected using the respective instrument response (to velocity units of m/s), and the mean and linear trend were removed.

In the study by Geimer et al. (2020), Rainbow Bridge showed clear normal modes, although the higher-order modes are partly overlapping (Figure 1a, b). We include this arch in our study as an example of having well-defined modes. For Corona Arch (Figure 1c), the numerical models by Geimer et al. (2020) predicted two modes between 5 Hz and 6 Hz, but only

one single peak could clearly be observed in the experimental data (Figure 1d). Therefore, we include this arch as an example of having a possibly hidden mode. At Squint Arch, the opposite phenomenon was observed: while two peaks were observed in the power spectrum between 10 Hz and 15 Hz, only one mode was predicted by the numerical model in this frequency range (Figure 1e, f). Finally, the large array data set acquired at Musselman Arch provides the possibility to test the techniques presented here using a dense sensor array, highlighting the value of retained phase information.

**Table 1: Location, span, and data acquisition characteristics for each arch investigated (coordinates in WGS84).**

| Site | Span [m] | Latitude | Longitude | Sensors | Acquisition date | Duration | Highlighted attribute |
|------|----------|----------|-----------|---------|------------------|----------|-----------------------|
| Rainbow Bridge | 84 | 37.0775 | -110.9642 | 1 TC 20-s | 24 March 2015 | 3 hours | Clear modes, $f_2$ and $f_3$ overlapping |
| Corona Arch | 34 | 38.5800 | -109.6201 | 1 TC 20-s | 8 October 2017 | 1 hour | Hidden mode $f_3$ |
| Squint Arch | 12 | 38.6465 | -110.6739 | 1 TC 20-s | 1 February 2018 | 1 hour | Modes $f_1$ and $f_2$ overlapping |
| | | | | 6 Zland 5-Hz, nodes | 30 April 2018 | 2 hours,50 minutes | |
| Musselman Arch | 37 | 38.4359 | -109.7699 | 32 Zland 5-Hz nodes, arranged in two parallel lines | 14 February 2017 | 2 hours | Large array data set |

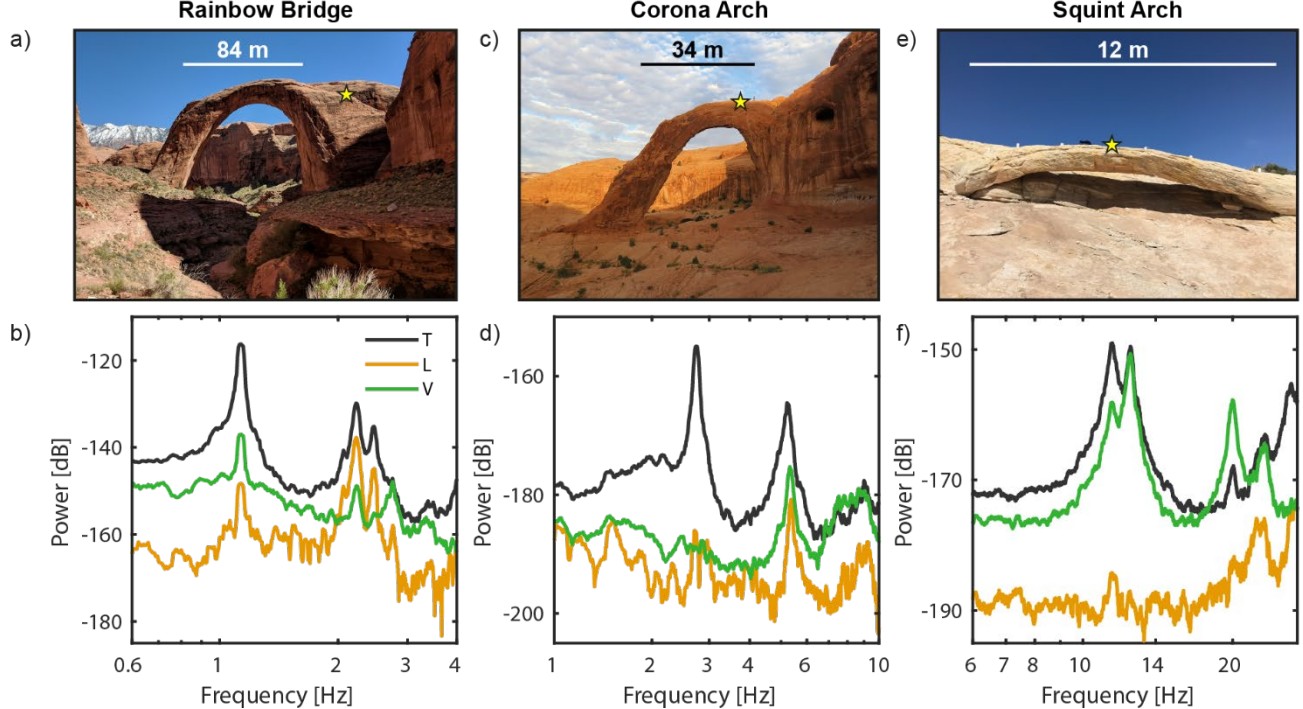

**Figure 1: a) Photograph of Rainbow Bridge with the sensor location marked by the yellow star. b) Power spectra recorded at Rainbow Bridge. Components are oriented transverse to the arch span (T), longitudinal or parallel to the arch (L) and vertical (V). Relative power is given in decibel [dB] units of spectral velocity [m²/s²]. c and d) Photograph and power spectra of Corona arch. e and f) Photograph and power spectra of Squint Arch. Photographs in panels a) and c) from Moore et al. (2020).**

## 3 Data processing

### 3.1 Peak-picking and polarization analysis

In previous studies of the dynamic response of natural rock arches, the resonant frequencies of the landform were determined by selecting the local maxima of the power spectra of the recordings, so-called "peak-picking" (Starr et al., 2015;Moore et al., 2018, e.g., $f_1$ in Figure 2a). The corresponding modal damping ratio can be estimated using the half-power bandwidth technique, where the frequencies left and right of the resonant frequency $f_n$ are selected ($f_A$ and $f_B$, respectively) as those where power has decreased to $1/\sqrt{2}$ (or approximately -3 dB, see Figure 2a). The damping ratio $\zeta$ is then obtained by

$$\zeta = \frac{f_B - f_A}{2 f_n} \ .$$ (1)

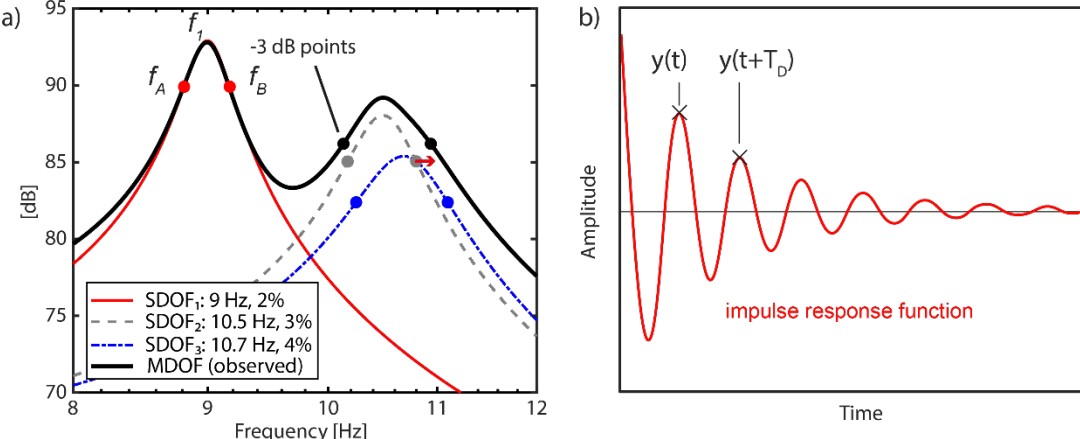

**Figure 2: a) Frequency response of three example single-degree-of-freedom (SDOF) systems with different damping ratios (%) and their superposition (a multi-degree-of-freedom-system, MDOF). Markers indicate the -3 dB or half-power points of each response curve, which are used to compute damping by the half-power bandwidth technique (see Equation 1). Note that mode 2 and 3 merge to one single mode bell, which causes an overestimation of modal damping (4.1% instead of 3.0%; indicated by red arrow). The third mode at 10.7 Hz cannot be observed (i.e., is hidden) in the MDOF power spectra. b) Impulse response function of a structure resulting, for example, from active excitation. Damping is determined from the logarithmic decrement technique (Equation 2) by measuring the amplitudes separated by one period.**

Mode shape information can be retrieved by polarization analysis (PA), for example, using the approach by Koper and Hawley (2010) as applied to rock arches by Moore et al. (2016) and Geimer et al. (2020). These single-station techniques are easy to use and provide reliable modal parameters in the case of well-separated modes. However, they fall short in case of closely spaced or overlapping modes, as the mode bells are not visible or are not corresponding to the underlying resonant mode (e.g., Papagiannopoulos and Hatzigeorgiou, 2011;Wang et al., 2012). This is illustrated for the example of three single-degree-of-freedom (SDOF, see Appendix A) systems illustrated in Figure 2a: one well-defined mode at 9 Hz is damped at 2%, whereas two closely-spaced modes at 10.5 Hz and 10.7 Hz are damped with 3% and 4%, respectively. The superposition of the three SDOF provides the resulting response of the multi-degree-of-freedom (MDOF) system (black line in Figure 2a), which is observed in the power spectrum. Analysis of the well-separated fundamental mode is straight forward, as the peak corresponds to the resonant frequency and applying the half-power bandwidth technique provides the correct damping ratio of 2%. In contrast, the peaks of both higher modes merge to one single mode bell at 10.5 Hz with an apparent damping estimate of 4.5%. Therefore, the superposition of the two modes results in broadening of the mode bell and consequently overestimation of damping. Furthermore, the third mode cannot be detected in the power spectra. In addition to damping overestimation by close and hidden modes, the half-power bandwidth technique tends to overestimate damping due to spectral leakage (Seybert, 1981).

The most direct estimate of modal damping ratios is obtained by active source experiments where the structure studied is excited artificially and energy dissipation is measured, for example, in the time domain by the logarithmic decrement $\delta$ (Figure 2b):

$$\delta = ln\left(\frac{y(t)}{y(t+T_D)}\right) = \frac{2\pi\zeta}{\sqrt{1-\zeta^2}} \qquad (2)$$

Here, $y$ represents the amplitude of the measured quantity (e.g., acceleration, velocity) at time $t$ and $T_D$ refers to the damped natural period (i.e., the inverse of the resonant frequency). For small damping (<20%; Chopra, 2015), this can be approximated and solved for $\zeta$:

$$\zeta \simeq \frac{\delta}{2\pi} \qquad (3)$$

Active source experiments can be considered to provide good estimates of damping ratios, but their application is restricted to structures that can be excited artificially (without inducing damage, Magalhães et al., 2010). In contrast, passive (i.e., ambient vibration) experiments can be applied on a broad range of structures but are subject to larger uncertainties (up to 20% is possible, e.g., Au et al., 2021;Döhler et al., 2013;Gersch, 1974;Griffith and Carne, 2007).

## 3.2 Enhanced Frequency Domain Decomposition

We processed three-component ambient vibration data using Enhanced Frequency Domain Decomposition (EFDD) (Brincker et al., 2001a;Brincker et al., 2001b;Brincker and Ventura, 2015;Michel et al., 2010). The method first computes the cross-power spectral density between all input traces and for every discrete frequency. Next, singular value decomposition for each frequency provides the singular values and singular vectors. The singular values can be understood as the collection of virtual SDOF systems of the structure, which enables detection of close and hidden modes that are not visible in the power

spectrum. The first singular value shows peaks at the dominant natural frequencies of the system. If present, overlapping secondary (i.e., non-dominant) modes result in elevated higher singular values. Resonant frequencies are then determined from analysis of the singular value plot, and the singular vector at the identified frequencies gives the three-dimensional modal vector (i.e., mode shape) of the chosen mode, with higher singular vectors representing the mode shape of non-dominating modes. These processing steps represent the Frequency Domain Decomposition method described by Brincker et al. (2001b).

The half-power bandwidth technique could now be applied on the singular values to estimate damping, as the bias by modal superposition is now addressed. However, spectral leakage may still broaden the mode bell.

A more accurate technique to estimate modal damping is the Enhanced FDD (EFDD) technique, introduced by Brincker et al. (2001a). Here the mode bell around each resonant frequency is picked manually and transformed to the time domain, providing the impulse response function (see Figure 2b). Energy decay in the linear part of the impulse response

function is expressed by the damping ratio $\zeta$, which can be determined using the logarithmic decrement technique. Linear regression of the zero-crossing times within the linear part of the decay curve additionally provides an updated estimate of the resonant frequency. The advantage of EFDD over the half-power bandwidth method is that the damping estimate is not based on only three picks but on a curve fitting approach, which reduces errors introduced by noise. However, EFDD still tends to

overestimate damping due to spectral leakage (e.g., Bajric et al., 2015). Detailed description of the EFDD processing workflow applied in this study can be found in Häusler et al. (2019) and Häusler et al. (2021), who applied the technique on unstable rock slopes.

### 3.3 Covariance-driven Stochastic Subspace Identification

The second technique used in this study is the Covariance-driven Stochastic Subspace Identification (SSI-COV) method (Peeters and De Roeck, 1999;Van Overschee and De Moor, 1993;van Overschee, 1996). Like EFDD, SSI-COV is a modal analysis technique frequently used in civil and mechanical engineering. Contrary to EFDD, SSI-COV is a time-domain parametric technique, which searches for the best set of modal parameters (resonant frequencies and modal damping) representing the observed structural response in a mathematical manner, i.e., minimizing misfit between modeled and observed data. Because it is a time-domain approach, overestimation of damping from spectral leakage is avoided. The most important processing parameter is the maximum lag time between two time samples used for computing the covariance matrices, which should be two to six times larger than the longest eigenperiod of the structure. Other user-controlled parameters include the number of possible modes, the accuracy threshold for modal frequency and damping, the maximum spectral distance inside a cluster, and the variation of the minimum modal assurance criterion (e.g., Allemang and Brown, 1982), which is a measure of the similarity of the mode shape at neighboring frequencies. We applied the SSI-COV algorithm software by Cheynet (2020), which is based on the implementation by Magalhães et al. (2009) and was used for comparison to EFDD on long suspension bridges (Cheynet et al., 2017). We followed the parameter suggestions by Cheynet (2020) and chose the parameters in a trial and error approach such that the first three resonant modes were reproduced (see Table A1). As SSI-COV establishes a mathematical model of the structure studied, the dynamic response can be defined by poles and zeros (in the sense of mathematics of complex numbers). Therefore, the term "pole" can be used as representative for "resonant mode" and is used hereafter to be in line with the terminology in the field. Since SSI-COV is a parametric method, its resulting resonant frequencies should be verified by a frequency-domain technique to prevent misinterpretation by model overfitting.

Results from SSI-COV (and other SSI variants) are illustrated using stability diagrams (e.g., Figure 3c). Initially, the structure's response is modeled with a low number of modes (poles), which is continuously increased to the maximum number of poles defined by the user. The maximum number of poles should be chosen to be significantly larger than the expected number of modes in order to establish an overdetermined mathematical model. The resulting resonant frequencies for each mode at every model run are plotted in the stability diagram (blue crosses in Figure 3c). Repeated poles, i.e., identical or very similar values for resonant frequencies, damping, and mode shape, represent stable poles and can be identified as vertical stacks of poles in the stability diagram (red circles in Figure 3c). Poles not fulfilling the user-defined accuracy criteria are not interpreted as stable poles and are scattered at arbitrary values as a result of noise fitting. Stable poles are clustered using hierarchical clustering, grouping poles with similar characteristics to the final resonant modes of the structure.

## 4 Results

We observe the first three resonant modes of Rainbow Bridge determined by the single-station measurement at 1.1, 2.2 Hz, and 2.5 Hz (Figure 3a-c). While the fundamental mode ($f_1$ at 1.1 Hz) is distinctly separated from other spectral peaks, the second and third modes ($f_2$, $f_3$) occur at closely spaced frequencies but are clearly identified by the elevated second singular value. Damping is estimated at between 0.6% and 1.3% for all three modes (Table 1). For the fundamental mode, we estimate damping at 0.9% and 0.6% using EFDD and SSI-COV, respectively, which is significantly lower than estimated by Geimer et al. (2020) using the half-power bandwidth method (2.4%). Modal vectors (i.e., azimuth and incidence angle with a lower hemisphere projection) derived by EFDD are similar to the PA results of Geimer et al. (2020) with some minor differences for $f_3$ and a 180° ambiguity in the azimuth of the nearly horizontally polarized mode $f_2$. Note that Geimer et al. (2020) allowed a polarity flip for mode shapes with sub-horizontal incidence angles equal to or larger than 85° in order to compare to numerical models. As SSI-COV and EFDD provide similar results, we only compare values from EFDD to PA in Table 2 and provide SSI-COV results in Table A2.

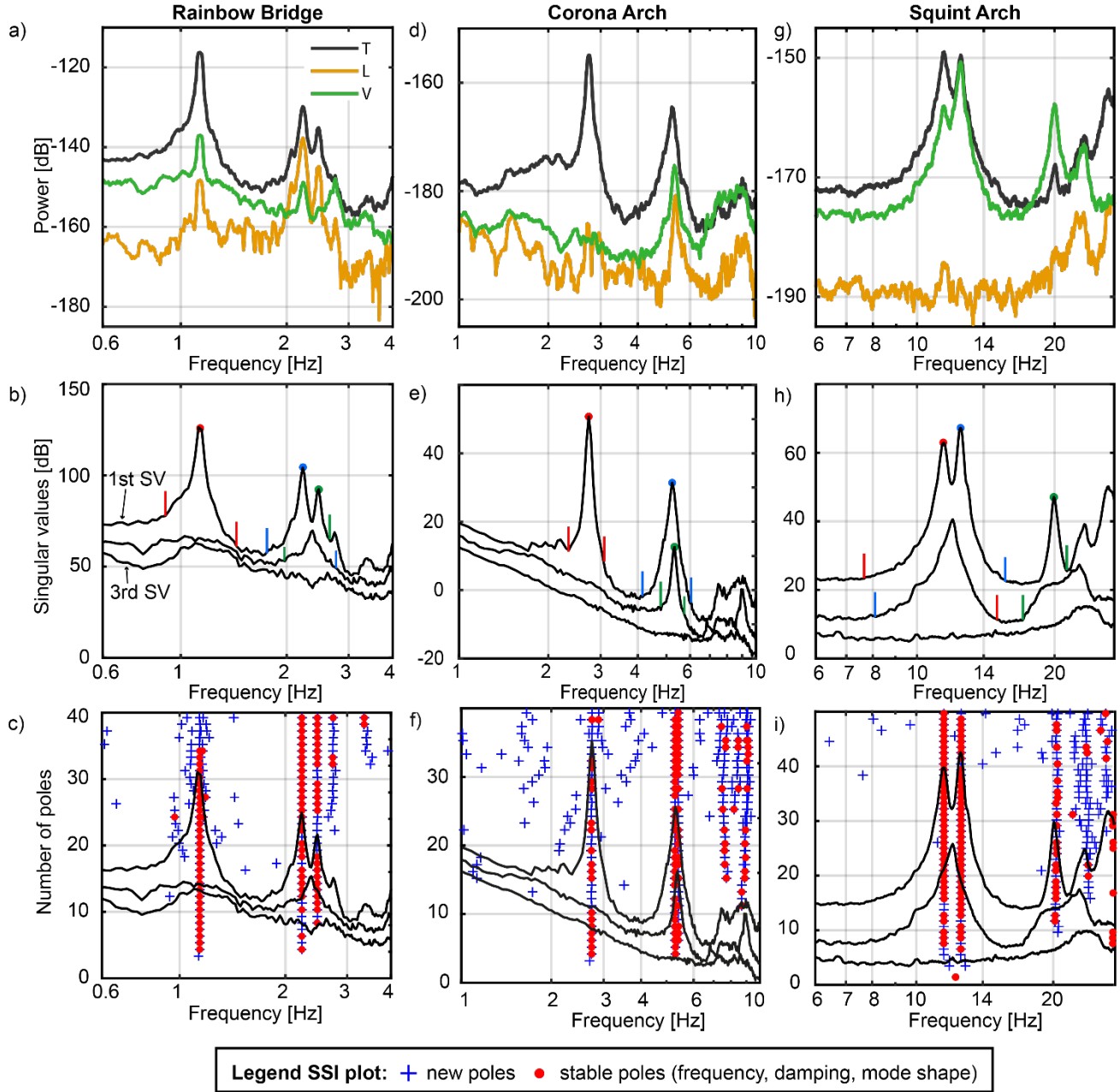

**Figure 3: a)** Power spectra of Rainbow Bridge, **b)** singular value (SV) plot of EFDD analysis at Rainbow Bridge. The largest SV always represents the first SV (top line). Solid markers indicate the resonant peaks, while vertical colored lines indicate the extent of the corresponding mode bell, **c)** singular value plot of Rainbow Bridge with SSI-COV poles superimposed with increasing number of poles. Each pole is marked with a blue cross, stable poles (in terms of resonant frequency, mode shape, and damping ratio) are marked with a red circle. Unstable poles (i.e. blue crosses at distance from stable poles) arise from noise fitting. Subplots **d)** to **f)** and **g)** to **i)** are the same as **a)** to **c)** for Corona Arch and Squint Arch, respectively.

The singular value plot of Corona Arch reveals two distinct spectral peaks at 2.7 Hz and 5.3 Hz (Figure 3e). However, the second singular value also peaks at ~5.3 Hz, indicating the presence of a closely spaced mode at that frequency. Therefore, we confirm the two close modes proposed by Geimer et al. (2020). However, EFDD and SSI-COV suggest nearly identical frequencies for $f_2$ and $f_3$ (5.3 Hz) while Geimer et al. (2020) selected more separated frequencies (5.0 Hz and 5.4 Hz) based on numerical modelling and PA. For EFDD, the mode shape vector of the non-dominant mode $f_3$ is determined by the second

singular vector. Modal vectors for $f_1$ and $f_2$ resolved by PA and EFDD are in good agreement, however azimuth and incidence differ for $f_3$. While EFDD and SSI-COV gave similar values for incidence of $f_3$ (44° and 54°, respectively), PA estimated incidence at 73°. Damping is estimated between 0.9% and 2.0% for all three modes, with 0.9% and 1.4% for the fundamental mode (via SSI-COV and EFDD, respectively). These values are again slightly lower than the half-power bandwidth estimates of Geimer et al. (2020, 1.9%). Damping ratios for $f_2$ and $f_3$ are between 1.5% and 2.0% with EFDD and SSI-COV providing

similar damping ratios within the expected uncertainty range.

For Squint Arch, we observe two closely spaced modes at 11.5 Hz and 12.5 Hz, and a third mode at 19.9 Hz (Figure 3h, i). Geimer et al. (2020) interpreted the first two peaks as one mode as it could not be confirmed as a separate mode by numerical models. Our analysis of the second mode suggests the modal vector has a steeper incidence angle compared to the first mode (49°), and is therefore oriented 60° from $f_1$. If the two spectral peaks were analyzed separately by PA, the match

between PA and EFDD is very good (see values in brackets in Table 2). The damping ratio of 1.6% determined by Geimer et al. (2020) is in perfect agreement with the estimation by SSI-COV but differs slightly from the EFDD result (2.4%).

**Table 2: Overview of resonant frequencies, modal damping ratios derived by EFDD and SSI-COV, and modal vectors (azimuth and incidence angle) estimated by EFDD and polarization analysis (PA) for Rainbow Bridge, Corona Arch, and Squint Arch. The values in brackets for Squint Arch are derived by PA if $f_1$ and $f_2$ were interpreted as separate modes (see supplementary information to Geimer et al., 2020). Incidence angle corresponds to the lower hemisphere projection.**

| | Frequency [Hz] | Damping EFDD [%] | Damping SSI-COV [%] | Azimuth EFDD [°] | Azimuth PA [°] | Incidence EFDD [°] | Incidence PA [°] |
|---|---|---|---|---|---|---|---|
| Rainbow Bridge | | | | | | | |
| Mode $f_1$ | 1.1 | 0.9 | 0.6 | 145 | 145 | 85 | 85 |
| Mode $f_2$ | 2.2 | 1.2 | 0.9 | 122 | 304 | 85 | 84 |
| Mode $f_3$ | 2.5 | 1.2 | 1.3 | 17 | 23 | 86 | 82 |
| Corona Arch | | | | | | | |
| Mode $f_1$ | 2.7 | 1.4 | 0.9 | 70 | 70 | 89 | 89 |
| Mode $f_2$ | 5.3 | 1.9 | 2.0 | 248 | 250 | 85 | 83 |
| Mode $f_3$ | 5.3 | 1.5 | 1.9 | 225 | 238 | 44 | 73 |
| Squint Arch | | | | | | | |
| Mode $f_1$ | 11.5 | 2.4 | 1.6 | 39 | 221 (39) | 71 | 61 (72) |
| Mode $f_2$ | 12.5 | 1.6 | 1.1 | 221 | n/a (221) | 49 | n/a (49) |
| Mode $f_3$ | 19.9 | 1.5 | 2.0 | 140 | 148 | 16 | 16 |

We demonstrate the ability of EFDD to retrieve the full-length normal mode shapes at Squint Arch, where data acquired by a nodal geophone array during a separate experiment are available (Figure 4b, raw power spectra are shown in Figure A1 a-c). We note that modal frequencies for $f_1$ and $f_2$ increased by about 1 Hz compared to the single-station measurement. No other higher modes are visible on the singular value plot during this measurement. Modal vectors for the first two modes at all stations resulting from EFDD analysis are shown in Figure 4c and d. We were not able to define a set of SSI-COV parameters that could successfully reproduce the observed modes.

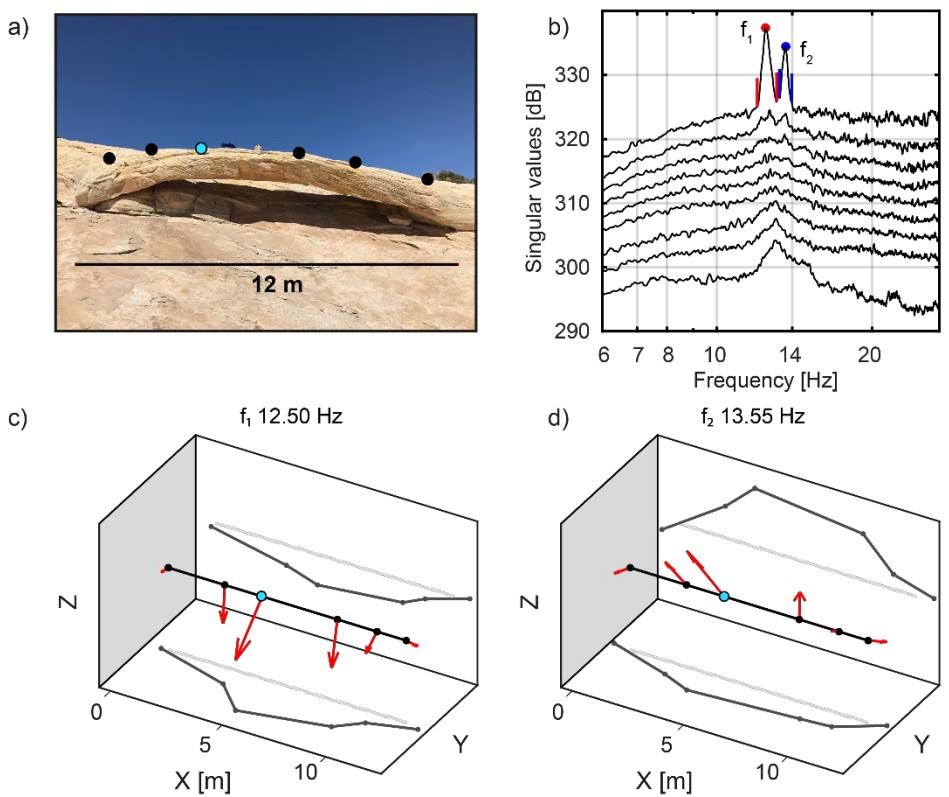


**Figure 4: Modal analysis of Squint Arch. a) Photograph of Squint Arch during the array measurements using nodal geophones, b) singular value plot of EFDD analysis showing the first nine singular values. Solid markers indicate the resonant peaks, while vertical colored lines indicate the extent of the corresponding mode bell, c) Projections of the mode shape at 12.5 Hz onto the Y-Z and Y-X plane. d) Projections of the mode shape at 13.6 Hz. Mode shapes are normalized to the station indicated by a cyan marker. The array**
**geometry in panels c and d (black dots and line) is simplified for illustration purpose.**

We performed EFDD and SSI-COV modal analyses on geophone array data acquired at Musselman Arch revealing the first four resonant modes at 3.4 Hz, 4.2 Hz, 5.6 Hz, and 6.6 Hz (Figure 5b, c; raw power spectra are shown in Figure A1 d-f). The resonant frequencies and mode shapes are in good agreement with results of the single-component cross-correlation analysis by Geimer et al. (2020). Visualization of the three-dimensional modal vectors for each station determined by the first
singular vector derived by EFDD are shown in Figure 5d-g. The first two modes are full-span, first-order bending modes in the horizontal and vertical directions, respectively. The third mode is a nearly symmetric second-order vertical bending mode with node point at the center of the arch. Mode four is a slightly asymmetric second-order horizontal bending mode with node point shifted towards the eastern abutment. Modal damping ratios for the first three modes are estimated at 1.3%, 1.0%, and 1.9% with EFDD and 1.3%, 1.1%, and 1.6% with SSI-COV. Note that the second singular value is elevated at each of the
resonant frequencies. However, in the case of Musselman Arch, this phenomenon is caused by an anomalous sensor component and not by close or hidden modes, as discussed in Section 5.1.

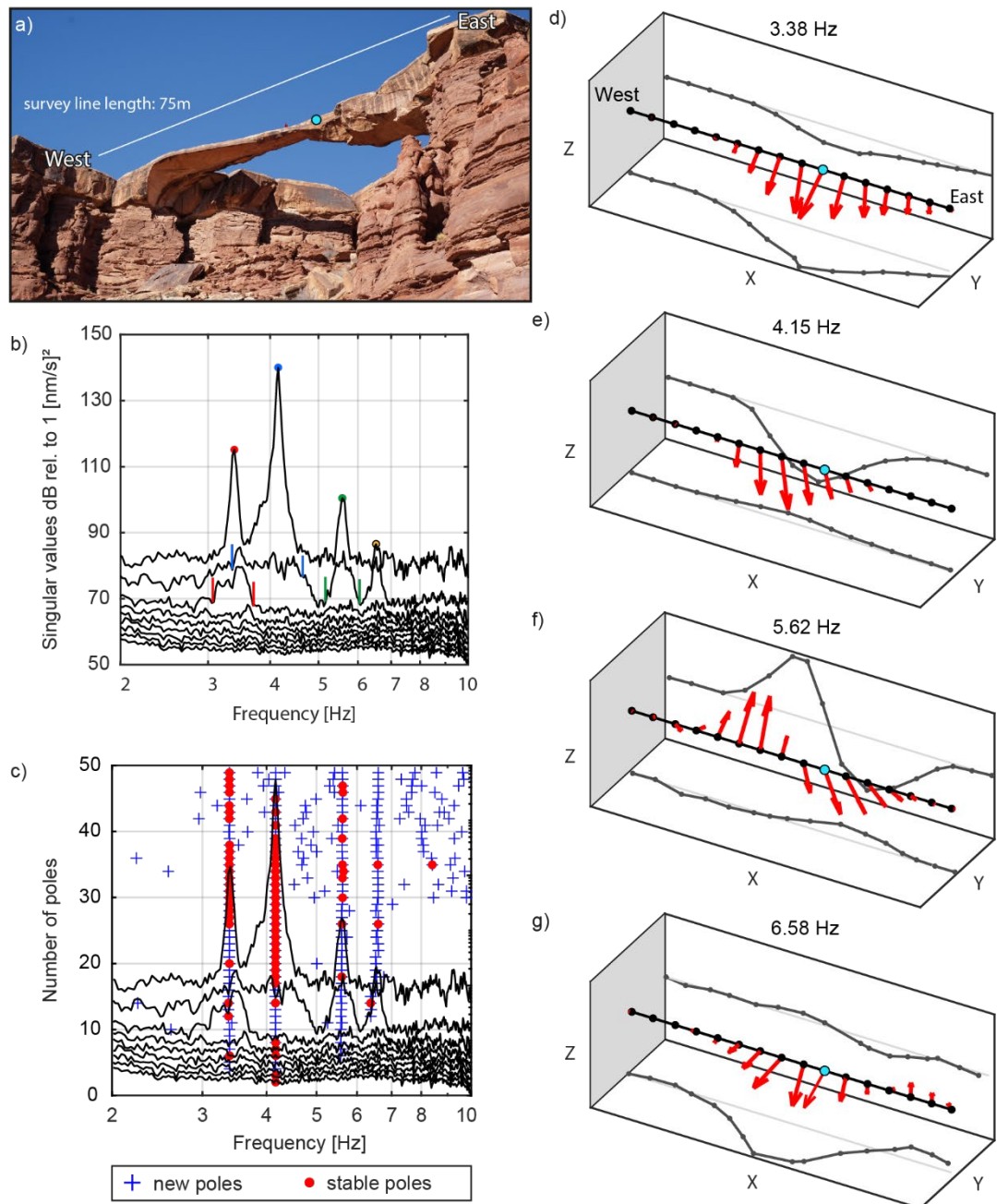

**Figure 5: a) Photograph of Musselman Arch, b) first nine singular values of the EFDD analysis. Solid markers indicate the resonant peaks, while vertical colored lines indicate the extent of the corresponding mode bell used for damping estimation, c) singular value plot with SSI-COV poles superimposed: stable frequency (blue cross) and stable mode shape (red circle). d) to g): 3D normal mode shapes at the first four resonant frequencies (3.38 Hz, 4.15 Hz, 5.62 Hz, and 6.58 Hz) with projections onto the X-Z and X-Y planes. The mode shapes are normalized to the reference indicated by the cyan marker. For better visibility, only one of the two parallel geophone lines is displayed, while the mode shapes of both lines are comparable (see Figure A2). Photograph panel a) by Kathryn Vollinger.**

## 5 Discussion

### 5.1 Modal identification

At Rainbow Bridge, the first three spectral peaks are well separated. Therefore, simple peak-picking on the power spectra provides the same resonant frequencies as obtained by EFDD and SSI-COV. All techniques provide comparable values for azimuth and incidence of the modal vector. The largest discrepancies between EFDD / SSI-COV and PA is 6° and 4° for the azimuth and incidence, respectively. These discrepancies are in the range of the misfit between field observations and the numerical models presented by Geimer et al. (2020). Therefore, we conclude that, for well-separated modes, all techniques provide similar results within the range of uncertainties.

At Corona Arch, only two resonant peaks are observed in the power spectra between 1 Hz and 7 Hz. The first includes both horizontal components, while the second also includes the vertical component. The numerical model by Geimer et al. (2020) predicted two close modes at the location of the second peak, which supported their interpretation of a close hidden mode. However, the same model also showed significant misfit between model prediction and observed data. With EFDD and SSI-COV, the presence of two close modes can be verified (note the elevated second singular value in Figure 4e). For the example of Corona Arch, we demonstrate that EFDD and SSI-COV are valuable techniques to detect close and hidden modes.

At Squint Arch, the power spectra show two resonant peaks between 10 Hz and 14 Hz, both including all three components. However, the numerical models by Geimer et al. (2020) only predicted one mode. Consequently, they interpreted this doublet-peak as the signature of one mode alone. In contrast, EFDD and SSI-COV independently indicate two closely-spaced modes. Therefore, these two techniques help to identify the resonant modes at Squint Arch.

At Musselman Arch, all four resonant modes between 1 Hz and 10 Hz are observed in the power spectra, as the resonant frequencies are well separated. However, the large array dataset results in 3 x 32 power spectra to be analyzed, which is an extensive effort (Figure A1). In addition, the power spectra do not provide direct evidence if the peaks correspond to one single mode or if there are additional close modes. Here, EFDD combines all input traces in one single plot, providing a direct illustration of the resonant modes and indicating that no close modes are present. In addition, the mode shapes can be directly plotted by evaluating the singular vectors. Therefore, the EFDD analysis of the large Musselman Arch dataset provides a demonstration of the user-friendliness and simplicity of the EFDD technique.

We observed that the second singular value at Musselman Arch is elevated to about 85 dB at each of the resonant peaks, which corresponds to the value of the first singular value at frequencies where no resonance occurs (Figure 6a). The difference in the overall noise floor, i.e., to the level where the singular values remain flat (~70 dB), is about 15 dB. Such nearly constant gaps between singular values across a broad frequency range is indicative of an erroneous sensor component or sensor coupling issue, which increases the noise floor to the noise level of the faulty component (e.g., Brincker, 2014). In our survey, the anomalous component was identified to be the transverse horizontal component of sensor 000, which is evident from the elevated power spectra in Figure A1d and the anomalous mode shape vector in Figures A2g. By omitting sensor 000 in our EFDD analysis, the overall noise floor drops to 70 dB and removes the elevated higher singular values at the third and

fourth resonant frequency (Figure 6b). The second singular value is still elevated between the first two resonant modes,

indicating that these modes have an overlapping bandwidth. In contrast to Musselman Arch, the higher singular values at Corona Arch were only elevated at the second resonant frequency and not at the first. In addition, the peak at the second singular value ($f_3$) exceeds the noise floor of the system, i.e., power of the first singular value outside the mode bells. This shows that the hidden mode $f_3$ at Corona Arch is indeed a structural mode and not an artifact induced by an erroneous sensor component.

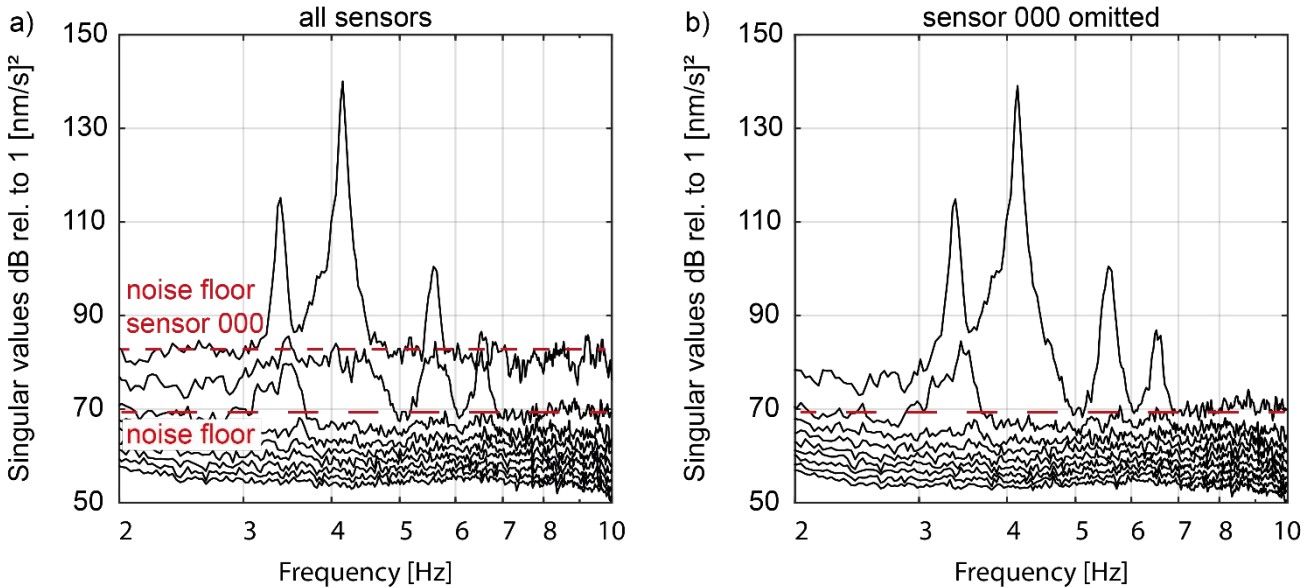


**Figure 6: First nine singular values of the EFDD analysis of Musselman Arch. a) Singular values including all 32 sensors. An anomalous horizontal component of sensor 000 increased the noise floor close to the peak of the fourth resonant frequency. However, the higher singular values are still clearly pointing out the four resonant frequencies. b) Singular values excluding sensor 000, resulting in a lower noise floor at 70 dB and clear peaks on the first singular value at the resonant frequencies.**

**5.2 Damping estimates**

For most resonant modes, EFDD and SSI-COV provide comparable damping results within the anticipated range of uncertainty. However, we observe that EFDD results in 30% to 35% higher damping ratios for the fundamental modes of Rainbow Bridge, Corona Arch and Squint Arch, as well as for the first higher mode of Squint Arch. We interpret this observation as an effect of damping overestimation through broadening of the resonant peak, caused by spectral leakage (e.g.,

Bajric et al., 2015).

Damping ratios obtained by the half-power bandwidth technique at Rainbow Bridge and Corona Arch are 75% and 53% higher than those estimated by SSI-COV, and 63% and 26% higher than estimated by EFDD. This is illustrated in Figure 7, where we compare damping ratios obtained by the various techniques for each arch. The resonant frequency and damping ratio derived are used to model a SDOF system, which is superimposed on the singular value plots. The amplitude

of the modeled SDOF is normalized to the maximum amplitude of the first singular value. The mode bell of the fundamental

mode of Rainbow Bridge is well reproduced by EFDD and SSI-COV, but damping is overestimated by the half-power bandwidth technique (Figure 7a).

Rainbow Bridge had the lowest fundamental frequency in the study by Geimer et al. (2020), who used the same settings to compute the power spectra for all arches, including arches with higher resonant frequencies such as Squint Arch.
Therefore, it is likely that these parameter settings were not ideal to resolve the low resonant frequencies of Rainbow Bridge with sufficient resolution. Therefore, we interpret the discrepancy between the half-power bandwidth and EFDD and SSI-COV as a result of strong spectral leakage for the half-power bandwidth technique. However, neither SSI-COV nor EFDD is able to perfectly reproduce the mode bell due to its slightly asymmetric shape, likely reflecting the oversimplified assumption of a SDOF system. At Corona Arch, SSI-COV is capable of reproducing the mode bell of the fundamental mode and the half-
power bandwidth again overestimates damping (Figure 7b). SSI-COV and the half-power bandwidth technique yielded identical damping ratios for the fundamental mode of Squint Arch (Figure 7c). For higher order modes, the discrepancy between EFDD and SSI-COV is smaller on all arches.

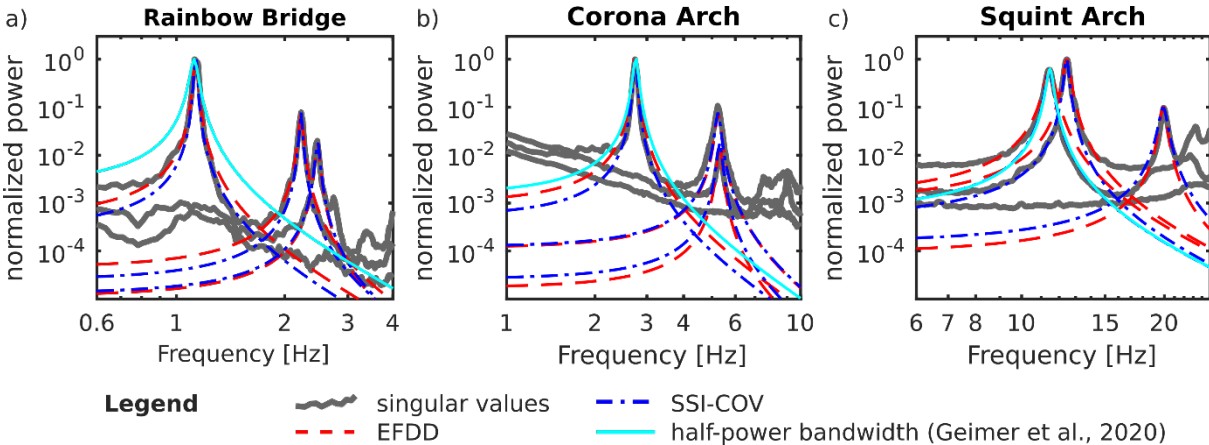

Figure 7: Singular value plots with spectra of single-degree-of-freedom systems modeled by using input data from EFDD (red), SSI-
COV (blue), and half-power bandwidth technique (cyan). At Rainbow Bridge (a) and Corona Arch (b), damping estimated by the half-power bandwidth technique is overestimated and does not match the mode bells, while the fit is good at Squint Arch (c).

## 5.3 Differences between surveys at Squint Arch

The normal mode analysis of Squint Arch resulted in different resonant frequencies for the single-station broadband measurement in February 2018 and the geophone array measurement in April 2018. We attribute this shift in frequency to
seasonal variations and mainly temperature differences (11.5°C for the single-station and 16°C for the nodal measurement, respectively, see also Starr et al., 2015). Seasonal effects are also expected to influence modal damping ratios and mode shapes (Häusler et al., 2021). Another difference between the two surveys is that only two modes can be detected by the geophone array. This is likely an effect of the higher self-noise of the Zland geophones compared to the broadband TC 20-s seismometer, which may be greater than the excitation level of the higher modes (e.g., Brincker and Larsen, 2007). We were also not able
to find a set of SSI-COV parameters that could reliably reproduce the resonant frequencies, which is again attributed to the lower signal-to-noise ratio of the geophone data (e.g., Brincker, 2014;Liu et al., 2019).

## 6 Conclusions

We applied Enhanced Frequency Domain Decomposition (EFDD) and Covariance-driven Stochastic Subspace Identification (SSI-COV) modal analyses on a set of four natural rock arches previously analyzed by Geimer et al. (2020) using frequency-dependent polarization analysis (PA). Our results show that EFDD and SSI-COV are able to determine the natural frequencies, damping ratios, and mode shapes of these landforms, including close, hidden, and higher resonant modes. For well-separated resonant modes, these techniques reproduce the results by Geimer et al. (2020). In the case of more complex spectra, EFDD and SSI-COV are able to extract additional modal details not resolved with PA. EFDD facilitated identification and interpretation of closely spaced (i.e., spectrally overlapping) and hidden modes at Corona and Squint arches. EFDD additionally combines information for all input traces in a single plot allowing rapid analysis of the dynamic response, especially when compared to picking the resonant peaks and determining polarization information on every station spectrum individually. The singular vectors resulting from EFDD can be directly interpreted as the three-dimensional modal deflection vector at each station, providing rapid and convenient visualization of mode shapes.

While modal analysis via peak-picking and subsequent PA has been shown to be satisfactory for adequately spaced spectral peaks and strongly amplified resonant frequencies, here we demonstrate that more sophisticated modal analysis techniques can provide refined modal characterization for more complex dynamic systems. Improving the accuracy and our understanding of resonance properties could in turn help generate more refined numerical models, facilitating more accurate arch stability assessment. Future efforts in modelling the dynamic response of rock arches (and other geological features) should additionally involve calibration of the modal damping ratio, as we have shown this parameter can be measured on complex structures.

Our results encourage widespread application of EFDD and SSI-COV modal analysis techniques, which are commonly used in civil engineering, to complement existing seismological techniques for dynamic analysis of geological features. Both techniques are well-suited for future near real-time monitoring of the structural integrity of geological features beyond rock arches: for example, rock slope instabilities, unstable glaciers, and freestanding rock towers. EFDD and SSI-COV are only two methods out of many other available algorithms for modal analysis, including other SSI variants and Curve Fit FDD (Peeters and De Roeck, 2001;Jacobsen, 2008). Therefore, future research could explore the potential of these techniques for applications involving modal analyses and monitoring of Earth surface landforms.

## Appendix A

The frequency response $H(\omega)$ of a single-degree-of-freedom (SDOF) system is given by

$$H(\omega) = \frac{1}{1 - \left(\frac{\omega}{\omega_n}\right)^2 + 2i\zeta\left(\frac{\omega}{\omega_n}\right)} \tag{A1}$$

with $\omega$ being the angular frequency and $\omega_n$ being the angular resonant frequency. $\zeta$ refers to the modal damping ratio and $i$ is the imaginary unit (see, for example, Chopra, 2015).

**Table A1: SSI-COV input parameters as defined by Cheynet (2020). Ts: time lag for covariance calculation (two to six times the natural period), Nmin: minimal number of model order, Nmax: maximum number of model order, ε cluster: maximal distance inside each cluster. Frequency accuracy (ε frequency), MAC accuracy (ε MAC) and damping accuracy (ε zeta) are set to 0.01, 0.05 and 0.04 for all analyses, respectively. The band-pass filter was chosen such that the resonant peaks observed in the spectra are included. Nmax and ε cluster were testes in a trial and error approach to obtain stable poles that match the first three observed resonant modes.**

| Structure | Ts [s] | Nmin | Nmax | ε cluster | Pass band [Hz] |
|---|---|---|---|---|---|
| Rainbow Bridge | 2.8 | 2 | 40 | 0.1 | 0.5 to 6 |
| Corona Arch | 1.2 | 2 | 40 | 0.15 | 0.8 to 12 |
| Squint Arch | 0.2 | 2 | 60 | 0.1 | 4 to 40 |
| Musselman Arch | 1.1 | 2 | 50 | 0.5 | 1 to 20 |

**Table A2: Modal parameters obtained by SSI-COV. Azimuth values labelled with an asterisk (\*) show a 180° ambiguity compared to EFDD and polarization analysis.**

| | Frequency [Hz] | Damping [%] | Azimuth [°] | Incidence [°] |
|---|---|---|---|---|
| **Rainbow Bridge** | | | | |
| Mode $f_1$ | 1.1 | 0.6 | 145 | 85 |
| Mode $f_2$ | 2.2 | 0.9 | 122 | 84 |
| Mode $f_3$ | 2.5 | 1.3 | 197* | 86 |
| **Corona Arch** | | | | |
| Mode $f_1$ | 2.7 | 0.9 | 69 | 90 |
| Mode $f_2$ | 5.2 | 2.0 | 72* | 87 |
| Mode $f_3$ | 5.3 | 1.9 | 43* | 54 |
| **Squint Arch** | | | | |
| Mode $f_1$ | 11.4 | 1.6 | 219* | 71 |
| Mode $f_2$ | 12.4 | 1.1 | 40* | 48 |
| Mode $f_3$ | 19.9 | 2.0 | 143 | 16 |
| **Musselman Arch** | | | | |
| Mode $f_1$ | 3.4 | 1.3 | n/a | n/a |
| Mode $f_2$ | 4.2 | 1.1 | n/a | n/a |
| Mode $f_3$ | 5.6 | 1.6 | n/a | n/a |

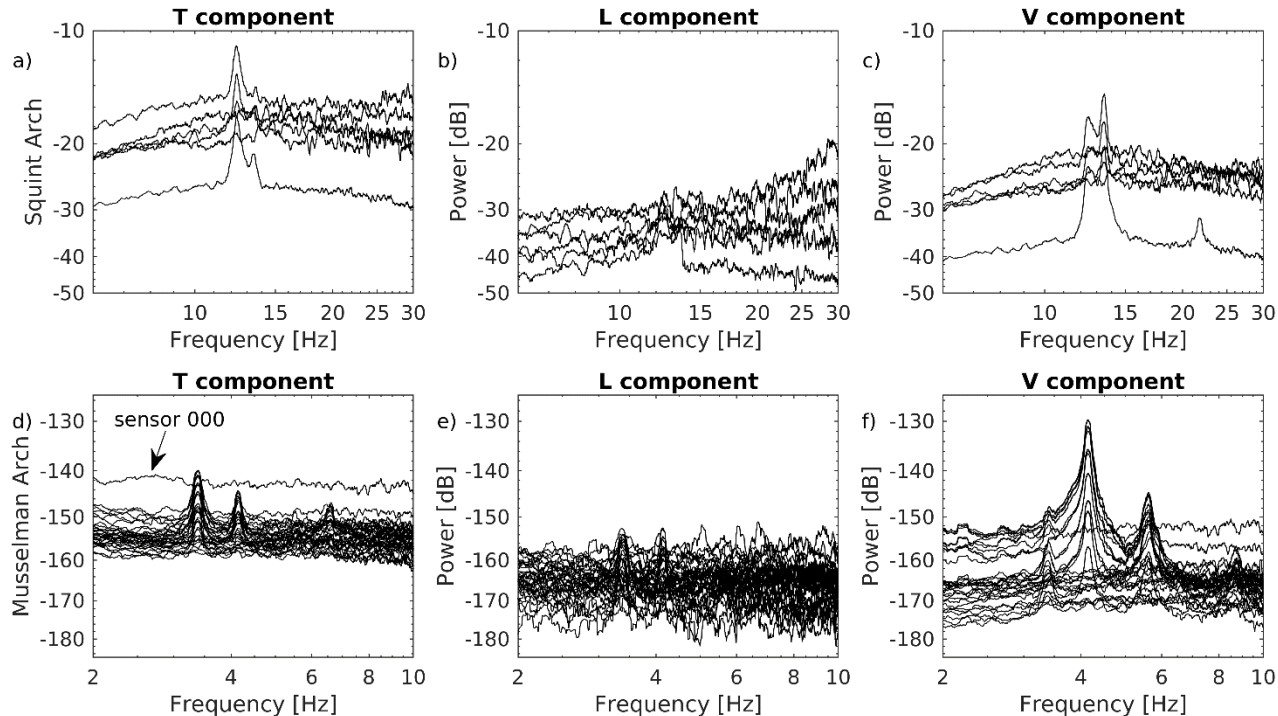

**Figure A1: a-c)** Power spectra of the geophone array deployed at Squint Arch, consisting of six sensors and three spatial recording components. Components are oriented transverse to the arch span (T), longitudinal or parallel to the arch (L) and vertical (V). **d-f)** Power spectra of the geophone array deployed at Musselman Arch, consisting of 32 sensors and three spatial recording components (T, L, V).

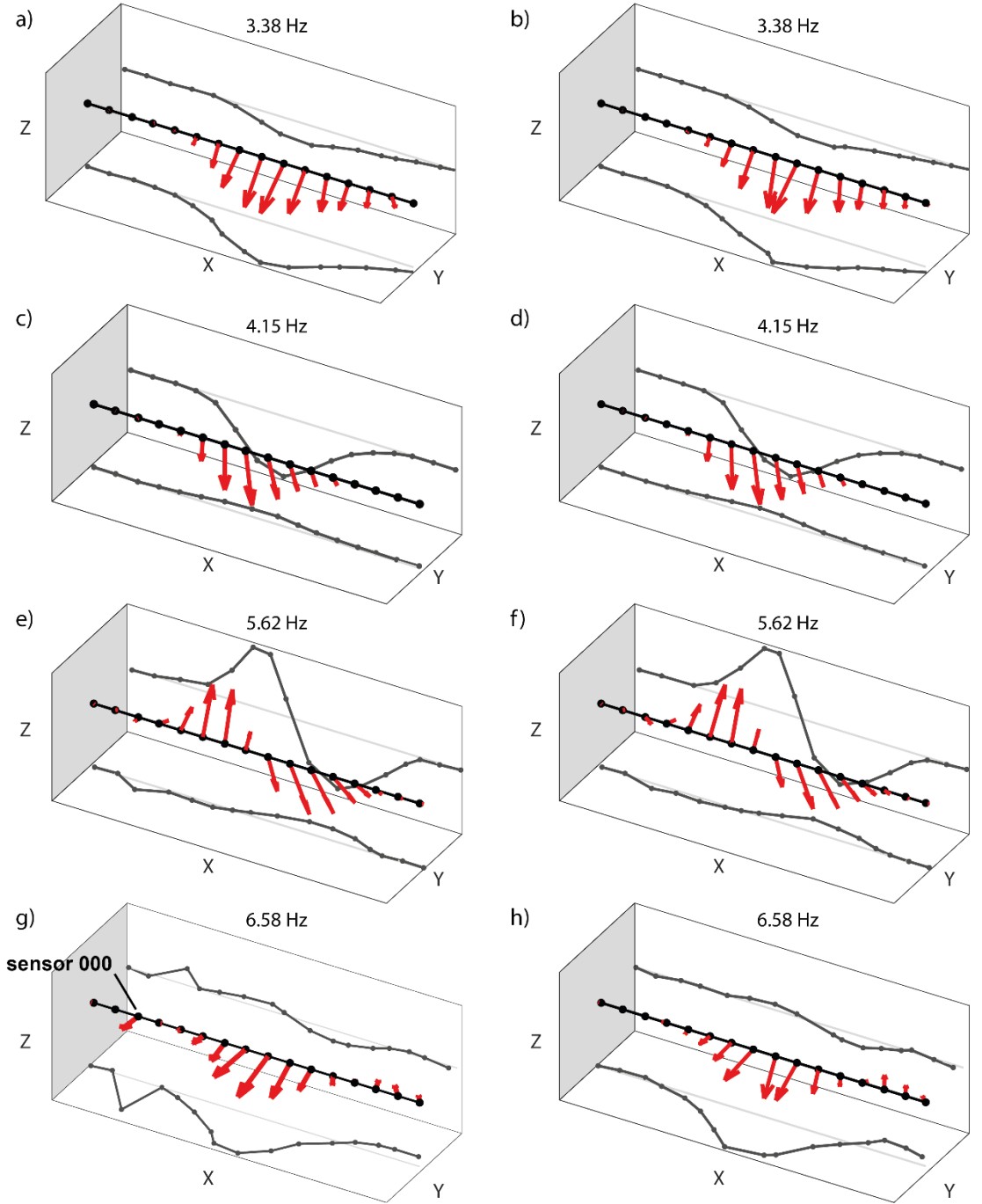

**Figure A2: 3D mode shapes of Musselman Arch for the two parallel lines of geophones and projections onto the X-Y and X-Z planes. The geophone line shown on the right is shown in Figure 5d-g in the main article. The mode shapes of the parallel lines are nearly identical.**

## Data availability

Data used in this study and originating from the study by Geimer et al. (2020) are available at [doi.org/doi:10.7278/S50D-G31E-NFW2](doi.org/doi:10.7278/S50D-G31E-NFW2). Data of the nodal deployment at Squint Arch are available at [https://doi.org/10.7914/SN/5P_2013](https://doi.org/10.7914/SN/5P_2013).

## Author contribution statement

The manuscript was written by MH with significant contributions from all co-authors. PG, RF, and JM acquired seismic data. MH carried out data processing and software development of EFDD modal analysis. PG performed data curation and validation of results. All authors reviewed and approved the manuscript.

## Competing interests

The authors declare that they have no conflict of interest.

## Acknowledgements

Support for this study was provided by the US National Science Foundation Grant EAR-1424896. M. Häusler was financed by ETH project grant 0-20361-17.

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
