# Peer review of "An Update on Techniques to Assess Normal Mode Behavior of Rock Arches by Ambient Vibrations"

_Earth Surface Dynamics, 2021_

## Author Response (AR1)

**Responses to esurf-2021-36**

**GENERAL COMMENTS**

Dear Editors
Dear Reviewers,

thank you very much for your thorough review and detailed feedback. We acknowledge that our manuscript appears very technical and introduces signal processing methods not often used in geoscience. Nevertheless, the geoscience community started in the past decade with normal mode analysis of geological features at the Earth's surface. Such analyses are common in mechanical and civil engineering, where advanced normal mode analysis tools have been in development for decades. These tools (e.g., the presented EFDD and SSI-COV) are standard methods which can also be obtained in commercial and open software packages. Therefore, this paper is not about introducing these techniques and assessing their uncertainties, as this was done by a large community over the past 20 to 30 years. This is why we originally decided to submit a "Short Communication".

The goal of this paper is to share our experience with these techniques and to motivate people doing normal mode analysis of geological structures to have a look at these techniques and benefit from the advances in other fields (as we did).

However, we realized that we did not succeed in explaining the limitations of simpler modal analysis techniques and why more advanced techniques could help to overcome these limitations. We also see that the manuscript needs some more explanations to be attractive for the broad readership of *ESurf*. Therefore, we significantly modified the introduction and method sections by including basic formulations from structural dynamics. We also added a separate section for a detailed discussion of the results.

However, we did not add a separate uncertainty analysis for the two techniques as we are convinced that this is sufficiently discussed in the literature. A key point is that the simpler techniques just fail in identifying and measuring closely-spaced and hidden normal modes. Therefore, these more advanced techniques enable the detection of these modes. The precision of these detections and estimates is not really the essence, because identifying a mode is apparently better than missing it. A repetition of the full mathematics of the two techniques, including their uncertainties, would result in a review article on normal mode analysis techniques. Such a review would probably not help to motivate geoscientists to apply these techniques (and we feel misses the scope of *ESurf*).

We still think that our manuscript has a "Short Communication" character, as we are applying established routines to available and published datasets, which might not be perceived as a complete piece of research. However, due to the increased length of the article (~6000 words, 25 pages), 6 figures and the inclusion of some fundamental concepts of structural dynamics, we suggest to continue with a normal manuscript and not with a "Short Communication". However, we are open for either format.

**REVIEWER 1**

*Häusler et al. present a study in which they apply two vibration analysis techniques to rock arches which are assumed to provide more detailed and robust results than previously employed methods. The study covers a relevant, timely and sufficiently novel topic, thus providing originality and an appropriate scope regarding the audience of the journal. The manuscript is in general of adequate quality; the language is scientifically correct and appropriate. Tables and figures are of proper quality with the exception of some too short captions (see below). It is good to see that the data is being made available. With some revision, it will make a valuable addition to the journal.*

*I see a few general concerns that I think need to be addressed, though. First, the study does not provide any benchmark data, but only relies on comparisons of the results of the two new techniques (EFDD and SSI-COV) – either amongst each other or with respect to results by Geimer et al. (2020). Thus, it is not possible to judge the overall quality/correctness of the presented modal information beyond that relative level of comparison. How can we know that the finer resolved results by EFDD and SSI-COV are real, due to the rock structure, and not just artefacts of either the data collection or the utilised methods? Perhaps this standing question can be solved by citing and discussing existing literature examples that provide the theoretical justification in this respect.*

→ Certainly, a true benchmark can only be achieved by very simple structural or numerical models, where simple analytical solutions are available. For more complex structures, a numerical counterpart cannot always be seen as a benchmark, as not all physical properties of the structure studied are known or the degree of details cannot be implemented in the model. This is especially true for modal damping, since the observed damping ratio is just an equivalent damping term summarizing all energy dissipating effects. To date, defining and quantifying the different physical damping mechanisms in structures is still an unrealistic task. Currently, the numerical models for natural rock arches are simplified and cannot serve as a benchmark to evaluate normal mode techniques, especially because the models themselves were calibrated by using the output of normal mode analyses – peak-picking technique, in case of Moore et al. (2018) and Geimer et al. (2020).

→ However, we see that we did not describe the advantages of the two presented techniques detailed enough and that a reader might wonder, why they should be superior. Therefore, in the revised manuscript, we highlight that both techniques are part of the standard procedure for normal mode analysis in civil and mechanical engineering since more than 20 years and that a broad range of studies exists, which compared these techniques with each other and to numerical models and provide the theoretical foundations. We added a number of these references to our manuscript and give some theoretical explanations on why simpler techniques fail to retrieve modal properties of close and hidden modes. We would like to highlight that we did not invent the two presented techniques or that we developed something superior. Our intention is to provide a bridging paper to facilitate access to engineering techniques for geomorphologists and geologists working on such landforms.

→ In terms of damping, a certain "benchmark" can be achieved by active experiments, i.e., by exciting the structure artificially and measuring the energy dissipation. For one of the arches

(Squint Arch), such an active experiment dataset exists. We extended the data processing section to explain this active experiment.

*Second, the data are not interpreted in a geoscientific way, hence the implications of the identified frequency modes for the landform. Are the discovered values in agreement with what one would expect for these landform geometries, rock types, stress distributions and environmental settings? I suggest the authors spend a few sentences on establishing the context of their analysis and the journal's main scope: fostering understanding of Earth surface dynamics. This is especially relevant when considering the pitch given in the introduction.*

→ This comment largely agrees with the issues raised by the editors before the (revised) manuscript was sent to review. Therefore, we hope that this issue was resolved in the manuscript that was finally posted for discussion (see https://doi.org/10.5194/esurf-2021-36 and https://doi.org/10.5194/esurf-2021-36-RC2). For example, we established a link to the stress distribution analyses by Moore et al. (2018) and explained how this analysis is integrated in the rock arch stability assessment (significant modifications in the introduction).

*Third, I see some ambiguities and arbitrarities in the presentation of the methods. It is good to see that the authors mention the multitude of model parameters but then, we simply get a reference to a table in the appendix in which the used parameter values are listed. The problem here is that these parameter values need to be introduced and justified. This should include a discussion of expected ranges, for example based on what other researchers have found or used. This may also include a description of the process that lead to the decision on the ultimate parameter values used. Currently, we have to take the parameter combination at face value, which is a fair bit from transparency and reproducibility.*

→ We mainly follow the default values given in the software by Cheynet (2020). We tried a few parameter combinations for Nmax and ε cluster to obtain a good reproduction of the first three resonant modes, as we now describe in the text and in the appendix.

→ The reader can perform a normal modal analysis by using the default values and modifying the time lag for the covariance calculation (which is described in the manuscript). We think, introducing all parameters in detail would be beyond the scope of this manuscript as they are best described in the software manual and cited literature. A detailed mathematical description of the SSI method would result in several pages of mathematical formulas, which would probably be out of the scope of *ESurf*. We think readers interested in the mathematics can refer to the cited textbooks.

→ We also believe that the exact values are not crucial for the message of the manuscript, which is to motivate geoscientists to use normal mode processing tools when doing normal modal analysis. There are many other scientific and commercial software available that are based on different parameters (open and proprietary, including EFDD and various versions of SSI, e.g., http://www.openmodal.com, https://svibs.com/artemis-modal/). We added a sentence to the conclusions to point at other SSI variants.

*Fourth, there is a mix of methods, results and discussion in each of the respective sections, which should be resolved. I give detailed comments to this issue further below. And I may emphasise that this is not a crucial flaw but one that should simply be resolved to give the manuscript a clear and organised structure.*

→ We added two new sections: section "Methods" is split to "Data acquisition and study sites" and "Data processing". The new section "Discussion" includes now all interpretation and discussions. As a consequence, the sections "Results" and "Conclusions" were shortened.

**Remark:** In the following, we provide point-by-point answers to your comments. As the line numbers refer to the (unpublished) first version of the manuscript, we also add the lines of the corresponding part in the official pre-print in parentheses. References to the manuscript officially published as a pre-print are named "pre-print version", whereas references to the original manuscript, which was (accidentally) sent to you for revision is labelled "original manuscript".

*l.1 (1), The title is very (if not too) long and it also reads very (if not too) technical, with a lot of quite specific jargon, especially when considering the main scope and readership of the journal. I recommend to shorten the title, remove the detailed technical/methodological terms and in exchange to add more emphasis on the environmental context (e.g. "monitoring rock arch material strength evolution", but this is just a non-ideal example that I give to reveal what I might expect as a reader to see, feel free to adjust as you please).*

→ Agreed.
→ Modified to "An Update on Techniques to Assess Normal Mode Behavior of Rock Arches by Ambient Vibrations"
→ Note our general comment to remove the "Short Communication"

*l. 9-10 (9-10), you could also consider motivating the study by a geotechnical pitch, instead of or in addition to the hazard one. Especially since you do not discuss the hazard perspective in the interpretation section, at all.*

→ In contrast to the original manuscript, we picked up the hazard perspective in the "Discussions and Conclusions" of the pre-print version.
→ The pre-print version also includes a link to numerical models, which might be improved by the presented techniques.

*l. 27-29 (30-32), you need to better motivate this sentence/abstract. There is a break in logic, here. I suggest you first motivate by the needs to monitor the stability of these landforms. Then you briefly mention the classically used techniques and their shortcomings. Then, this gives you the pitch to identify the research gap and thus motivate the seismic approach as complementary solution.*

→ Changed as suggested.

*l. 36 (39), "resonant frequencies arise primarily due to changes in rock mass stiffness". This is true but there are also other important factors that control the frequency. See for example Bottelin et al. (2013) or the full story told by Lévy et al. (2010). These other important aspects should be mentioned, as well.*

→ Resonant frequencies are indeed solely governed by mass and stiffness (and damping, in case damping is large). Therefore, the factors described by Bottelin et al. (2013) and Lévy et al. (2010) are also acting on the stiffness.
→ But we agree that environmental effects have significant effects on the frequency and added a statement with this regard.

*l. 41 (44), "more invasive monitoring techniques", mention these other techniques, perhaps in relation to my comment above (l. 28-29).*

→ We primarily mean taking in-situ rock samples for laboratory tests.
→ This is now stated explicitly and the sentence is moved to the first paragraph of the introduction.

*l. 81 (101), can you explain why this refers to overlapping modes? Is this specific to the method?*

→ Normal modes with similar resonance frequencies might overlap, i.e. they are not forming two separate peaks in the power spectrum. Here, EFDD is a straightforward technique to detect the presence of such overlapping modes, because it decomposes the response into a set of single-degree-of-freedom systems. The number of peaks on higher singular values directly shows the number of modes present in a given frequency band.
→ We modified the sentence to show that overlapping modes do not always exist, but might be present.

*l. 84 (104), "picked manually", here some information must be given on the criteria used to define these manual picks.*

→ There is a variety of possible approaches for choosing the mode bell. For clear mode bells, as we observe it on the rock arches, one chooses the frequency band which visually corresponds to the mode bell (i.e., has the "bell  shape").
→ For reproducibility, we marked the mode bells on the singular value plots in Figures 3 and 4.

*l. 92 (97+111), please define "output-only technique", this seems quite generic to me.*

→ "Output-only" is the technical term used in modal analysis to clarify that no information on the input signal is needed.
→ We removed the term, as it is not relevant in the context and causes confusion.

*l. 99 (119), You need to tell why each of the parameter values was chosen. Currently this is just arbitrary. It seems that the paremeter combinations determine to a significant extent the output of the technique, so this is a crucial part that deserves clear description and rigour. Have you tested different combinations and optimised them manually/iteratively? Have you used values published by other authors? Did you set the parameters just arbitrary?*

→ See reply to general comment.
→ It is a mixture of the points you mentioned, but we mainly followed the recommendations of the software manual.

*l. 103-105 (123-125), this would actually be much better suited for the introduction when you motivate the two techniques and want to convice the reader of their appropriateness. I suggest to move this to the introduction. It certainly does not match here in the results section.*

→ This sentence is written in the method section.
→ We now extended our introduction to better highlight the advantages of the techniques.

*l. 107 (127) "modelled with a low number of modes", "the maximum number", please mention what a low number is and what the concrete maximal numbers were, and more important give a justification for these numbers.*

→ The minimum and maximum number of modes are given in Table A1 of the Appendix.
→ SSI-COV establishes a mathematical model to describe the dynamic properties of the structure. The number of modes describes the number of modes that are inserted in this mathematical model. The maximum number of modeled modes is higher than the physical models in order to have a mathematically over-determined system of equations (Peeters & De Roeck, 1999). However, every model run needs computational time. Therefore, the number of modes needs to be limited.
→ We added a sentence for clarification.

*l. 111 (131), "user-defined accuracy criteria", what are these criteria? Please describe and justify them.*

→ See reply to general comment.
→ The accuracy criteria are given in Table A1 of the Appendix (cluster, MAC and damping accuracy)
→ We added slightly more detail to explain the parameters. However, we are mainly using default values of the software.

*l. 116 (135 and following), actually I would like to see the "raw" data, in this case the spectrograms or spectra, if just to be convinced that from these raw products one cannot already see the same frequency modes as in the advanced analysis.*

→ We now also added the power spectra of the three seismometer components of each arch in a new Figure 1 together with the photographs of the arches as well as in Figure 3. For the array data, we added the spectra component-wise to the appendix. We think that showing 96 separate power spectra for Musselman Arch or 18 power spectral for Squint Arch would not be beneficial.
→ In addition, we added short discussions of these raw spectra in the introduction to show the challenges for each site.

*l. 126 (150), Fig. 2, it might help to colour code the singular value lines to indicate which is the first, second, third SV (a legend would be needed in that case, too). Also, this figure should contain the PSDs of the data sets to compare the new outcomes against them. In panels c, f and i, it remains elusive to me when a pole is defined stable and when not. Is there any criterion that was used? For example in c around 2.7 Hz there are many apparently stable values that still are plotted as blue crosses.*

→ In FDD analysis, there are both philosophies, to color the singular value curves or not. We think that coloring them introduces a visual bias and prefer to stick to the uniform black lines for the singular values. This is especially important for close modes, where the SDOF systems involve multiple eigenvalues.
→ The first singular value is the line with the highest energy. By definition of the singular value, the lines never cross and are thus always separated. We added more detail to Figure 3 and its caption to explain the singular value plot.
→ Power spectra are added (see previous comment).

→ Yes, SSI-COV needs some stability criteria. These are introduced in the section 3 (Data processing). Their values are given in Table A2 of the Appendix.

→ We agree with your observation regarding the mode around 2.7 Hz. There is indeed a good number of stable poles and given the peak on the first singular value, we would indeed interpret this as a resonance frequency. The mode is weaker than the first three modes, leading to fewer "stable poles" compared to the other modes. However, we did not discuss this mode as we are focusing on the first three modes of every arch (except Musselman) to maintain readability and to keep the study comparable to the one by Geimer et al. (2020).

*l. 133-151, there are many occurences of interpretations in this part. Please separate presentation of results and their intepretation throughout the text. Here is just an example.*

→ We introduced a new section "Discussion" for discussions and interpretations.

→ However, at this particular location, we do not see much interpretation. We see the comparison to the work by Geimer et al. (2020) as part of the results, summarized in Table 1. If there are differences between the two studies, we now state them in the results section but discuss potential reasons in the discussion section.

*l. 141 (161), what means "good agreement"? Please quantify or leave it.*

→ Uncertainties of 20 to 30% for damping ratios can be considered as normal (see modifications and added references in the introduction). Therefore, with "good agreement", we mean an agreement within this expected range.

→ Modification for clarification.

*l. 159-163 (179-181), This approach has not been mentioned in the methods. Please move it to the methods section and also give more context and information on SDOF.*

→ We added the introduction of SDOF to the "Data processing" section and give the basic formulation of a SDOF system in the appendix.

*l. 163 (184), "half-power bandwidth technique", here as well this may be better mentioned in the methods section (or introduction if it is more appropriate there) but in any way, some short explanation of the term and its implication needs to be added, especially in a non-seismologist journal.*

→ Description and formulation of the half-power bandwidth technique is now added to the "Data processing" section.

→ We moved the entire discussion on damping to the new section "Discussion".

*l. 176-185, this section is also full of repeated interpretations of results. Please separate these materials into the appropriate sections, "Results" and "Discussion".*

→ We now separate the section of "Discussions and Conclusions" in two separate sections and move the discussions on damping previously written under "Results" to the "Discussions".

*l. 189/Fig. 3, the cation is too short and gives too little context about the presented material of this figure.*

→ We expanded the captions.

*l. 206-208 (224-226), this is repetitive and redundant. Consider removing.*

→ Yes, it is repetitive. However, we believe that a conclusions should start with a short repetition of the main task and goal of a study.

*l. 208 (226), what means "well suited"? Can you quantify this? Besides, how do you know the methods are well suited if there is no independent benchmark data to compare against? Data like rock mechanical model predictions of expected frequencies and their degree of overlap? Ovreall, this will be tricky to show. See my general comment on this issue.*

→ Please refer to our general comment in the beginning, where we address this issue in detail.
→ In general, the good performance of EFDD and SSI is widely accepted in the field of modal analysis. In fact, they are standard techniques in the core fields of modal analysis (civil and mechanical engineering). We modified the introduction and method section to better explain this.
→ Our goal is not to prove the superiority of these techniques but to transfer these techniques to geoscience.

*l. 209-211 (229), How can we be sure these are not just artefacts but indeed emerging due to a "better" approach?*

→ This is related to the previous comment. The aim of this manuscript is not provide a proof that these two techniques are superior based on their methodology and advanced signal processing. This was shown in detailed technical papers in civil and mechanical engineering, which made these techniques state-of-the-art in these fields (in fact, there are already extended and improved versions of the methods used here). We now include a few additional references in the introduction.
→ With "additional modal detail" we specifically mean close and hidden modes. These modes can simply not be seen by solely looking at the power spectra.
→ We modified the sentence to highlight this point.

*l. 216-220 (234-238), these descriptions are not really an outcome/implication of this study but rather a generic property of the method that should be better mentioned in the introduction (or methods section).*

→ We now added a paragraph to the "Data processing" section to describe the "conventional" approach and show these descriptions already there.
→ We would still like to keep this repetition in the conclusions to have a reminder for the reader, why SSI-COV and EFDD might be useful to be applied.

**REVIEWER 2**

*This short communication of Hausler et al. presents the comparison of two classical methodologies currently used for operational modal analysis of engineering structures applied. It is an original and interesting idea to apply both methods to geomorphological features such as rock arches. The objective is clear, the paper can be fluently read and the results are interesting. Nevertheless I have several questions and remarks that I would like to be adressed/discussed by the authors before final publication.*

**General remarks**

*Need to exactly define what is EFDD and SSI-Cov methods. I understand is meant to be a short paper, but this are not current techniques in Geomorphology, so some hints will help readers a lot. Especially, how the damping is estimated by each method ? and what are the main processing steps in both.*

→ We agree that these are not very widespread techniques in geomorphology. However, there is a vast literature available, including studies in geomorphology (glaciers, landslides, rock towers, sedimentary valleys: Bottelin et al., 2013; Ermert et al., 2014; Häusler et al., 2019, 2021; Mercerat et al., 2021; Moore et al., 2019; Poggi et al., 2015; Preiswerk et al., 2019), which also provide more details on the techniques. We believe that interested readers with the intention to apply the methods can refer to the cited literature and that including the mathematical formulations to this manuscript would not add much value.

→ We added a paragraph on the estimation of damping, including the formulation of the logarithmic decrement and the half-power bandwidth technique and a figure to show damping in time as well as in frequency domain.

*Need to specify and discuss that only two of the four sites have array data. So, for example, what is the advantage to use advanced techniques in single station measurements ?*

→ We performed array measurements at Squint and Musselman arch, as described in Section 3 (Data processing)

→ The main advantages of EFDD and SSI-COV are the same for single-station and array data: the possibility to resolve close and hidden modes and that only one plot needs to be analyzed. We modified the manuscript at various points to better highlight this advantage.

→ We agree that the efficiency of EFDD to analyze the data and display the results are not substantial for a single-station analysis. However, the advantage to resolve close and hidden modes is still given.

**Specific Remarks**

Abstract.

*Line 14-16 "Therefore, we investigate two algorithms well-established in the field of civil engineering through application to a set of natural arches previously characterized using conventional seismological techniques."*

*-I would not call "algorithms" but instead "methods" for EFDD and Co-SSI.*

→ Changed to "technique".

*-Please specify what do you mean by "conventional seismological techniques" -> may be the polarization analysis (Lines 17-18) ?*

→ Yes, changed accordingly.

*Line 19: the authors state that the proposed advanced techniques have "the capability to resolve closely spaced modes and provide stable damping estimates" and provide "more detailed characterization of dynamic parameters". After reading the whole paper, I'm not convinced that the results presented validate both statements (unless dynamic parameters, the authors mean exclusively modal shapes and frequencies)*

→ We agree that our paper should not be about the stability assessment of modal damping ratios, as this is discussed in technical literature.
→ We show that EFDD and SSI-COV are able to retrieve damping estimates of the modes analyzed and that these estimates scatter in the expected range.
→ A key point is that EFDD and SSI-COV are able to detect close modes and estimate their damping ratio. We discuss the issue of uncertainties for damping estimations now when introducing the damping techniques (Data Processing) and in the new section "Discussions".

**Introduction**

*Line 71. EFDD is really "well-suited" for distinguish closely spaced modes ? Can the authors underline what enhanced EFDD is compared to FDD ?*

→ Yes, the detection of closely spaced modes was a key motivation for Brincker et al. (2001) to invent FDD. The ability of FDD to distinguish close modes is also described in US patent US6779404B1.
→ EFDD is the extension of FDD, which also includes the estimation of damping. FDD already provides the resonant frequencies and mode shapes. EFDD transforms the mode bell to time domain to retrieved the impulse response function and finally to obtain the modal damping ratio.
→ We clarified the difference in the "Data processing" section of the manuscript.

*Line 80-85. Sort of repetition of the main capabilities of each proposed advanced technique. Please delete.*

→ Deleted.

*Line 84. The authors suddenly include "rock slope instabilities", but they were not studied in the present work. Please clarify or delete.*

→ We believe that it is important to show that these techniques can be applied on other geomorphological structures.

→ We added the references to these studies.

**Methods**

*Line 85. The methods section begins with the site presentations and instrumentation. Please adapt the section's title. The authors should clearly specify why each site has been chosen for the present study. Different instrumentations have been applied (See my General Remark).*

→ Title changed to "Data acquisition and study sites"
→ We added a paragraph to explain why we chose these sites and added a figure showing the power spectra which should help to understand the issues with the different structures.
→ We now additionally include the sensor name in Table 1.

*Line 114. Please specify "data residuals" of what ? : Velocities, cross-correlation traces, at which sensor, components, etc.*

→ Simply spoken, it is the residuals between the model and the observed data in time-domain. The more elaborated answer would cover a few pages and can be found in Van Overschee and De Moor (1993) and Peeters and De Roeck (1999).
→ We are fully aware that SSI is not an easy mathematical concept. However, we think that our goal should be to show that SSI is a method to build a mathematical model that best fits the data. We simplified the sentenced to gain more clarity.
→ We believe, this short description is justified at this location, as the mathematical method itself is a standard technique in civil engineering since several decades. Furthermore, we gave all the references and the SSI-COV software we are using is broadcasted as open source by Cheynet (2020).

*Line 117. "Modes (i.e. Poles)" Need a MUCH longer explanation.*

→ Poles and zeros are terms in complex mathematics and used in structural and electrical engineering to define a system's transfer function. Resonant frequencies are defined at the poles (a singularity). In this context, it is fair to use them synonymously to the term "mode", as a basic introduction to the concept of complex numbers, poles and zeros, and transfer functions would be beyond the scope of the paper.
→ We added a sentence for clarification.

*Line 120-125. This paragraph fits better in the Introduction part : the fact that the two methods have been previously compared in other context.*

→ Moved.
→ In addition, we added a few references to show that both techniques are standard methods in civil engineering for structural analysis and to show their strengths and weaknesses.

*Line 130. This paragraph include technical details of the SSI that are not clearly followed by the reader. Please clarify*

→ Based on this comment, it is difficult to assess which parts are not clear.

→ We provide the references to the original literature and textbooks on the technique.

→ We slightly modified the paragraph.

**Results**

*Line 137-140. The authors state that damping values for the fundamental mode are quite different from the three techniques. By the way, the authors should previously define the "half-power bandwidth method" used by Geimer (2020), with respect to the "mode bell" fitting of EFDD.*

→ We now introduced the half-power bandwidth in the "Data processing" section.

→ Yes, we stated that damping is 0.9 and 0.6 % for EFDD and SSI-COV but 2.4 % for half-power bandwidth technique. The reason for this difference is discussed in the Discussion section.

→ We also added a statement that damping is very difficult to estimate, regardless of the technique. Uncertainties of 20 to 30% are not exceptional (e.g., Au et al., 2021; Döhler et al., 2013; Gersch, 1974; Griffith & Carne, 2007).

*Line 135. For this example of Rainbow Bridge, it may help the reader to recall that here a single station analysis is being used, and that is the reason a single Modal vector is compared.*

→ Changed as suggested.

*Line 153. Corona Arch. It seems here two closed modes are found between 5.0 and 5.4, but EFDD and SSI_Cov indicate exactly the same frequency ! So the advanced methodologies were not indicated to separate close modes ? Please rephrase the paragraph to explain this behavior.*

→ In fact, this is an excellent example for the performance of the two techniques: by just counting the number of spectral peaks between 5 and 6 Hz, one would retrieve one single mode. In contrast, both EFDD and SSI-COV recovered two modes. Therefore, the advanced techniques clearly show that there are two close modes, whereas simple peak picking (PP) would fail. This is exactly the demonstration of the strength of EFDD and SSI-COV.

→ We added a paragraph of discussion for each arch to the new section "Discussion".

*Line 158. The concept of "modal incidence" is not clear at all. Please redefine. In fact, could the authors propose other terminology (for the single station measurements) because it is quite confusing. I would not see an "incidence" angle for a mode. If I understand correctly, the authors would like to compare vector orientations in 3D, it is not better simply "azimuth, dip and rake" ?*

→ The incidence corresponds to the angle of the modal vector and the vertical axis. We use this term, as it was used in other studies in this context, for example Moore et al. (2019) and Finnegan et al. (2021), but especially by Geimer et al. (2020), who analyzed the same arches as presented in this manuscript. The term goes back to the original paper by Koper and Hawley (2010), who introduced the polarization analysis (PA) technique that was subsequently used by the aforementioned authors.

→ We acknowledge that "dip" would be a valid alternative for "incidence". However, we prefer to stick to the term "incidence" that is more established in the field.

*Line 160. here again damping values are different. Reasons ? Can the authors advance any uncertainty for each estimation ?*

→ We move the discussion of the differences in damping to the new "Discussion" section.

→ Estimating damping is indeed a challenging task and its uncertainties are much larger than for the estimation of resonance frequency. (e.g., Au et al., 2021; Döhler et al., 2013; Gersch, 1974; Griffith & Carne, 2007). Discussing these uncertainties in this manuscript would be beyond the scope of the paper. However, it follows from these studies that uncertainties of 20 to 30 % can be expected. We expanded the introduction of the different techniques and give references to the above mentioned publications.

→ In this light, the differences in damping between the techniques were expected. Cases, where larger differences were observed are now discussed in the new "Discussion" section.

*Line 164. Squint Arch. Mode splitting is proposed here for the two close modes at 11.5-12.5 Hz cause by anisotropy. I do not have access to the work of Geimer et al (2020) but it looks that "homogeneous numerical model" does not reproduce a mode-splitting phenomenon.*

→ Thanks for putting a finger on this. As the work by Geimer et al. (2020) is an important foundation but not open access. We uploaded this work on an institutional repository: https://geohazards.earth.utah.edu/images/grl60517_accepted.pdf.

*First I guess "homogeneous" should be replaced by "isotropic". In fact, it may be the case that heterogeneous (though isotropic) models may present these two modes with quite close frequencies, but completely different modal shapes. In fact, it seems to be the case from the EFDD results of the later experiment with the 6 node stations : the first one seems to be a longitudinal mode, while the second one seems to be bending in the transvere direction. Is that also confirmed by the single station analysis ( azimuth/incidence ) ? This should be discussed in the paper. In fact why looking to anisotropic models (rather complex) when may be a numerical modal analysis could support this two "close" modes ? It may be useful for the readers to get the Figures from Geiger et al (2020) co-author included in the present paper.*

→ Agreed, it should be 'isotropic'.

→ However, we removed the entire topic of the properties of the numerical model as this is not relevant for the scope of the manuscript.

→ Geimer et al. (2020) do not present a figure related to Squint Arch. However, the azimuth and incidence values presented in their paper are given in our manuscript in Table 2.

→ The single station analysis shows an incidence of 71° (19° off from horizontal) and 49° (41° off from horizontal) with an azimuth perpendicular to the arch. Therefore, both modes are first-order bending modes but with different vertical components. We observe the same behavior during both surveys. However, a direct comparison is not possible, as the modal properties might change between the surveys. We added sentence to the discussion and updated Figure 4 to better illustrate the mode shapes.

*I can not see why the full modal analysis with many sensors is not much exploited. For example, there is also the strange phenomenon of mode f3 (near 20 Hz) that completely dissapears in the second cam-*

*paign. It would be really helpful to compare the recordings from the broadband seismometer (1st cam-pagin) and the node exact (or closely) located node for the 2nd campaign. This would be quite useful for new planned operational modal analysis campaigns with node-type equipment.*

→ We agree that evaluating different sensor types is an interesting topic and is important for fu-ture modal analysis campaigns.

→ However, this manuscript has the goal to show advanced normal mode analysis techniques used in engineering to the geoscience community. Therefore, we prefer focusing on this topic, espe-cially as this was originally planned as a Short Communication. A comprehensive instrument evaluation would be a topic for an independent study.

→ As a reply to your comment, we show here the power spectra of the broadband of the first array and the closest nodal geophone. We attribute the differences in modal parameter to environ-mental effects (mainly temperature, see Starr et al., 2015). However, such effects are not the scope of this manuscript.

[Figure]

*Line 187. Puzzled about this interesting active experiment. More information needed.*

→ We added a sentence to describe the active experiment in the section "Data processing".

*Last thing, about Squint Arch. What would be the damping value estimated from the EFDD or SSI of the nodal campaign ? The peaks in the SVD looks quite different from the ones of Figure 2c). It will be useful to compare the two campaigns in light of different instruments, number of sensors, both for frequency and damping characterization. Which is the impact ?*

→ As described, we were not able to find a set of SSI parameters that leads to stable results. There-fore, we do not have estimates based on SSI.

→ EFDD provides damping ratios of 2.5 and 0.8 %. Obviously, there is a difference to the single-station measurements for the second mode. However, the data originates from two different campaigns with different environmental conditions, which most likely also caused the frequency shifts and can affect the mode shapes. This is now mentioned in the "Discussions".

→ We agree, that a study on the effects of instruments and number of sensors would be very interesting and could result in an experiment optimization study (optimum amount of sensors, find cost-effective but still reliable sensor, etc.). However, these questions are beyond the scope of our manuscript. The questions on instrumentation and number of sensors is independent from these algorithms. For your interest: there is a student report available that discusses some sensor-related differences on rock arches, though not the sensors used in this study: https://geo-hazards.earth.utah.edu/images/UROP_Final_Report_CR.pdf

*Lines 210-215. It is rather dissapointing that the experiment with the higher number of sensors (2x16) is not much further discussed (with respect to the other 3 cases, only one paragraph !). For example, two lines were measured: synchronously ? with a reference station ? how much time duration ? Were these the same instruments that the ones in Squint Arch ? Why a twisting mode (torsional mode) is not being identified ? What about the dimensions (especially width, thickness) of the arch ?*

→ Yes, both lines were acquired synchronously, the duration is given in Table 1 (column "duration"). We added a sentence for clarification.

→ Both sites were instrumented with Zland 5-Hz nodal geophones, as written in Section 2 (Data acquisition and study sites). We now include the sensor type also in Table 1 (column "Sensors").

→ Yes, the normal mode shapes are normalized to a station in the middle of the arch. Such normalization to a sensor with large modal deflection is common practice in modal analysis. We marked the reference station in the figures.

→ Indeed, we did not observe a torsional mode. Possible explanations for this include the mode appearing at a higher frequency which was not strongly excited above background noise levels, or an array geometry that was not sufficiently spaced to capture torsional motion. However, the objective of our manuscript is to present two processing techniques to assess rock arches. The investigation on why no torsional mode is identified or how this relates to the dimensions of the arch is beyond the scope of the paper. However, hopefully, this paper is encouraging for geologists and geo-structural engineers to use these techniques and find answers for these questions.

→ Regarding the extent of the paragraph: Musselman arch is a relatively easy case with no close or hidden modes. We use this site to demonstrate the ability of FDD to analyze all stations in one single plot (the singular value plot) and directly retrieve the mode shapes.

**Discussion and Conclusion**

*Line 227. I'm not fully convinced about the statement that both methods are "well-suited" to determine all dynamic parameters. Please rephrase, the objective of the paper was to look to differences in parameter determination. Anyway, the differences are important and the instruments for data acquisition seem to have much stronger impact than the methodology. Comment on that ?*

→ While PP and PA failed in recovering the closely-spaced mode (f2 and f3) at Corona arch and could not provide conclusive results at Squint arch regarding the two closely-space first modes,

EFDD and SSI-COV could both separate these close modes. Therefore, these two techniques are at least better suited than PP and PA, because they detect the modes and determine their modal parameters.

→ We showed that EFDD and SSI-COV can recover damping ratios also for hidden and close modes. In case of hidden and close modes, simple half-power bandwidth techniques on the individual power spectra cannot be applied, simply because the mode bell is not visible or cannot be identified as such. In addition, modal superposition leads to a broadening of the mode bell, as now illustrated in Figure 2a. However, we admit that the half-power bandwidth technique can also be applied on the singular values, which is still an improvement by FDD because it clearly shows the mode bell. We added a sentence to show this improvement.

→ We cannot follow the statement that the instruments have a larger impact than the processing techniques. The only arch where different sets of instruments were used in separate campaigns was Squint Arch. We compare our results to the study by Geimer et al. (2020), using the exact same data.

*Lines 235-240. Damping estimation (even with EFDD and SSI-Cov) is always difficult and I'm not convinced that the advanced techniques are more "robust" than the half-power bandwith picking. Is there no "spectral smoothing" in both advanced techniques ? In the EFDD a "mode bell" is fitted to an SVD singular value, then back transformed in time, and measured by logarithmic decay; and in SSI-Cov, as far as I understand, there is also a parameter fitting in the least-square sense. No smoothing and/or regularization at all ?*

→ Yes, we agree that damping is very difficult to estimate and that our manuscript should not focus too much on discussing robustness, especially since this is extensively discussed in more technical literature. We added some references on this topic (Au et al., 2021; Döhler et al., 2013; Gersch, 1974; Griffith & Carne, 2007).

→ SSI-COV is supposed to be more robust because there is no spectral smoothing (it is a time-domain technique).

→ A key element is that mode superposition leads to broadening of the mode bell, resulting in an overestimation of damping when analyzing the power spectra (as now illustrated in Figure 2). Therefore, EFDD is more accurate than the half-power bandwidth technique, (but not necessarily more precise).

→ In addition, EFDD (and other curve-fitting techniques) are considered to be more robust than half-power bandwidth because they fit a curve, i.e. many points, to the data, whereas the half-power bandwidth technique is based on three values only (resonance frequency plus the two -3dB points). Therefore, the half-power bandwidth technique is more sensitive to noise and outliers.

→ However, damping of close and hidden modes cannot be determined on power spectra, simply because the mode bell cannot be seen or identified as such. Therefore, we reformulate the paragraph in a way to show that these advanced techniques allow for estimating damping even of close modes. We refer to the literature for a discussion on the robustness of the techniques.

*In conclusion, I advise the authors to revise this paragraph, specially the statement concluding that more advanced techniques would give more robust estimates of damping. Robustness may only be assessed if a detailed uncertainty analysis is carried out: different time windows, spectral estimation, etc.*

→ See previous comment.

*Line 239. "determined by the active impulse measurement at Squint Arch." I'm really puzzled about this experience. There is not much information in the manuscript. If an active impulse was used (hammer?) , I could imagine relatively high frequencies involved. How damping at high frequencies can/may be compared with damping at the whole structure scale (freq < 15 Hz) ? I think the authors should give much more information of the active experience in the present paper.*

→ We added a sentence to describe the active experiment. The active experiments were performed in the study by Geimer et al. (2020). Here, we compare damping results of EFDD and SSI-COV to values published in that publication.
→ The arch was excited by stomping on the arch next to the ground (using a hammer on the arch or any other stronger sources would probably damage the structure, which must be avoided).
→ While there is certainly a source effect, this was minimized by applying a band-pass filter around the resonance frequency.

*Line 245-255. On the other hand, I agree with the authors about the capabilities of sensor arrays to better characterize modal shapes of different rock arches or geological structures compared to a single station approach.*

→ Definitively! We would like to point out again that here, FDD provides a very user friendly tool to retrieve the mode shapes and analyze the data in one single plot. Analyzing the 96 seismic traces recorded at Musselman arch with respect to resonance frequency, polarization, and damping is an enormous task. Single-station polarization analysis would still result in 32 individual analyzes.
→ We added a sentence to highlight this user-friendliness.

*Line 254. homogeneous "isotropic" models.*

→ Removed.

**REFERENCES**

Au, S.-K., Brownjohn, J. M. W., Li, B., & Raby, A. (2021). Understanding and managing identification uncertainty of close modes in operational modal analysis. *Mechanical Systems and Signal Processing, 147*, 107018. https://www.sciencedirect.com/science/article/pii/S0888327020304040

Bottelin, P., Lévy, C., Baillet, L., Jongmans, D., & Guéguen, P. (2013). Modal and thermal analysis of Les Arches unstable rock column (Vercors massif, French Alps). *Geophysical Journal International, 194*(2), 849-858. http://dx.doi.org/10.1093/gji/ggt046

Brincker, R., Zhang, L., & Andersen, P. (2001). Modal identification of output-only systems using frequency domain decomposition. *Smart Materials and Structures, 10*(3), 441-445. http://stacks.iop.org/0964-1726/10/i=3/a=303

Cheynet, E. (2020). Operational modal analysis with automated SSI-COV algorithm. *Zenodo*.

Döhler, M., Hille, F., Mevel, L., & Rücker, W. (2013). Estimation of modal parameters and their uncertainty bounds from subspace-based system identification. In K. Margit (Ed.), *IRIS Industrial Safety and Life Cycle Engineering - Technologies / Standards / Applications* (pp. 91-106): VCE.

Ermert, L., Poggi, V., Burjánek, J., & Fäh, D. (2014). Fundamental and higher two-dimensional resonance modes of an Alpine valley. *Geophysical Journal International, 198*(2), 795-811. https://doi.org/10.1093/gji/ggu072

Finnegan, R., Moore, J. R., & Geimer, P. R. (2021). Vibration of Natural Rock Arches and Towers Excited by Helicopter-Sourced Infrasound. *Earth Surf. Dynam. Discuss., 2021*, 1-21. https://esurf.copernicus.org/preprints/esurf-2021-43/

Geimer, P. R., Finnegan, R., & Moore, J. R. (2020). Sparse Ambient Resonance Measurements Reveal Dynamic Properties of Freestanding Rock Arches. *Geophysical Research Letters, 47*(9), e2020GL087239. https://doi.org/10.1029/2020GL087239

Gersch, W. (1974). On the achievable accuracy of structural system parameter estimates. *Journal of Sound and Vibration, 34*(1), 63-79. https://www.sciencedirect.com/science/article/pii/S0022460X7480355X

Griffith, D. T., & Carne, T. G. (2007). *Experimental Uncertainty Quantification of Modal Test Data* Paper presented at the 25th International Modal Analysis Conference, Orlando, FL, USA.

Häusler, M., Michel, C., Burjánek, J., & Fäh, D. (2019). Fracture Network Imaging on Rock Slope Instabilities Using Resonance Mode Analysis. *Geophysical Research Letters, 46*(12), 6497-6506. https://doi.org/10.1029/2019GL083201

Häusler, M., Michel, C., Burjánek, J., & Fäh, D. (2021). Monitoring the Preonzo rock slope instability using resonance mode analysis. *Journal of Geophysical Research: Earth Surface, n/a*(n/a), e2020JF005709. https://doi.org/10.1029/2020JF005709. https://doi.org/10.1029/2020JF005709

Koper, K. D., & Hawley, V. L. (2010). Frequency dependent polarization analysis of ambient seismic noise recorded at a broadband seismometer in the central United States. *Earthquake Science, 23*(5), 439-447. https://doi.org/10.1007/s11589-010-0743-5

Lévy, C., Baillet, L., Jongmans, D., Mourot, P., & Hantz, D. (2010). Dynamic response of the Chamousset rock column (Western Alps, France). *Journal of Geophysical Research: Earth Surface, 115*(F4). https://agupubs.onlinelibrary.wiley.com/doi/abs/10.1029/2009JF001606

Mercerat, E. D., Payeur, J. B., Bertrand, E., Malascrabes, M., Pernoud, M., & Chamberland, Y. (2021). Deciphering the dynamics of a heterogeneous sea cliff using ambient vibrations: case study of the Sutta-Rocca overhang (southern Corsica, France). *Geophysical Journal International, 224*(2), 813-824. https://doi.org/10.1093/gji/ggaa465

Moore, J. R., Geimer, P. R., Finnegan, R., & Michel, C. (2019). Dynamic Analysis of a Large Freestanding Rock Tower (Castleton Tower, Utah)Short Note. *Bulletin of the Seismological Society of America*. https://doi.org/10.1785/0120190118

Moore, J. R., Geimer, P. R., Finnegan, R., & Thorne, M. S. (2018). Use of Seismic Resonance Measurements to Determine the Elastic Modulus of Freestanding Rock Masses. *Rock Mechanics and Rock Engineering, 51*(12), 3937-3944. https://doi.org/10.1007/s00603-018-1554-6

Peeters, B., & De Roeck, G. (1999). Reference-based stochastic subspace identification for output-only modal analysis. *Mechanical Systems and Signal Processing, 13*(6), 855-878. http://www.sciencedirect.com/science/article/pii/S0888327099912499

Poggi, V., Ermert, L., Burjanek, J., Michel, C., & Fäh, D. (2015). Modal analysis of 2-D sedimentary basin from frequency domain decomposition of ambient vibration array recordings. *Geophysical Journal International, 200*(1), 615-626. https://dx.doi.org/10.1093/gji/ggu420

Preiswerk, L. E., Michel, C., Walter, F., & Fäh, D. (2019). Effects of geometry on the seismic wavefield of Alpine glaciers. *Annals of Glaciology, 60*(79), 112-124. https://www.cambridge.org/core/article/effects-of-geometry-on-the-seismic-wavefield-of-alpine-glaciers/8A3E41F662877AF916A32C5A7AD39298

Starr, A. M., Moore, J. R., & Thorne, M. S. (2015). Ambient resonance of Mesa Arch, Canyonlands National Park, Utah. *Geophysical Research Letters, 42*(16), 6696-6702. https://agupubs.onlinelibrary.wiley.com/doi/abs/10.1002/2015GL064917

Van Overschee, P., & De Moor, B. (1993). Subspace algorithms for the stochastic identification problem. *Automatica, 29*(3), 649-660. https://www.sciencedirect.com/science/article/pii/000510989390061W

---

## Author Response (AR2)

**Responses to esurf-2021-36**

**GENERAL COMMENTS**

Dear Editors
Dear Reviewers,

thank you very much for your review and the positive feedback. Please find our point-by-point replies to your comments in this document.

**REVIEWER 1**

*I see all my previous comments addressed or adequate explanations why they were not addressed. The quality, clarity and structure of the text improved and I see only a few minor formatting/structuring things that should or could be improved. I am thus very happy with the manuscript at its current stage and the elaborated reply letter by the authors. I believe the clarifications have significantly increased the readability and potential impact of the material.*

*Section 3.1 reads a bit like an introduction. Would it be possible to either move the introductory parts to the peak picking and polarisation approach to the introduction or rewrite the section so that it is more focused on the investigated landforms instead of too broad?*

- ➔ We added a few sentences to introduce the peak-picking approach already in the introduction.
- ➔ However, we would prefer to keep the rather extensive introduction and explanation of damping in Section 3.1, as we think that most readers are not familiar with structural damping and the techniques used to determine this parameter.

*Figure 3 would benefit substantially from adding the names of the landforms on top of a, d an g, so that one realises that each landform is represented column-wise.*

- ➔ Implemented.

*Figure 4, would it be possible/meaningful to also add the vector arrows (as in fig. 5 d-g) to the XYZ plots? This would solve the small consistency issue.*

- ➔ Yes, we now use the same style as used for Figure 5.

*Regarding the overarching discussion of the "Short Communication" format, I understand these points. Indeed, the clarifications have resolved the apparent ambiguity of introducing a new versus applying an established technique. As mentioned in the reply letter, the updated version did indeed do an excellent job in resolving this initial misunderstanding. As for the correct title convention, I would leave this decision to the editor.*

➔   Regarding the length of the article and the introduction of some more basic physical concepts, we still see this manuscript as a full article. However, we are absolutely open to the editor's decision and would also agree with a short communication.

**REVIEWER 2**

*I acknowledge the great amount of work the authors have accomplished to reorganize the paper and answer many of reviewer's remarks. The paper has substantially gained in clarity by separating Methods, Results and Discussion. Anyway, I still have some few points that I demand to be fully addressed before final publication.*

*1) Rainbow Bridge : for the fundamental mode at 1.1Hz, damping of 0.9/0.6 % EFDD/SSI-COV against 2.4% by Geimer et al (2020) by half-power fitting technique. In principle, this mode is not mixture of two close modes. Any idea of the reason behind this over-estimation ?*

➔   The half-power bandwidth technique is (like EFDD) suffering from spectral leakage. In case of different spectral resolution, also different (apparent) damping values are expected.
➔   We added the following explanation to the manuscript:
➔   "Rainbow Bridge had the lowest fundamental frequency in the study by Geimer et al. (2020), who used the same settings to compute the power spectra for all arches, including arches with higher resonant frequencies such as Squint Arch. Therefore, it is likely that these parameter settings were not ideal to resolve the low resonant frequencies of Rainbow Bridge with sufficient resolution. Therefore, we interpret the discrepancy between the half-power bandwidth and EFDD and SSI-COV as a result of strong spectral leakage for the half-power bandwidth technique."

*2) Corona Arch : interesting finding of the hidden mode around 5.3 Hz. On the contrary, nothing is explicitly said about the technique to identify it : marked peak in the 2nd Singular Value ? Please clarify, may be in the 3.2 section when presenting the EFDD ?*

➔   Exactly, the peak on the second singular value is picked and the mode shape of the 'hidden' mode is determined by the second singular vector.
➔   We added sentence for explanation.

*3) Squint Arch : do not agree with "EFDD to retrieve the full normal mode shapes" as the authors could only recognize the first two normal modes (data quality of the nodal geophone array). Please qualify your statement.*

➔   We mean "full-length" across the arch span, in contrast to a single station measurement.
➔   We modified to "full-length"

*4) Musselman Arch: from Fig 5b (EFDD technique) it seems that the 4th mode at 6.58 Hz is peaked in the 2nd singular value (SV). Did the authors extract the modal shape from the 2nd eigenvector ? Please clarify.*

➔ No, all resonant modes are picked on the first singular value and all mode shapes are based on the first singular vector.

➔ We specified this in the revised version.

➔ See next comment for more details.

*Additionally, the 3rd mode at 5.62 Hz is also present in the 2nd SV. The authors stated that there are clearly not "hidden" modes in this case. May the authors comment on this ? especially in relation with the hidden mode recognized in Corona Arch (from a peak in the 2nd singular value)*

➔ This is a very good observation, thank you for pointing this out. We agree that this deserves some explanation.

➔ In case of Musselman Arch, this signature on the second singular value is related to an anomalous sensor component, which increases the noise floor across the entire spectrum. Note that the second singular value always peaks at the noise floor. We cannot state with certainty whether the anomaly was caused by bad sensor coupling or by a technical issue.

➔ We could simply remove the erroneous sensor from the analysis. However, we think that discussing this issue is more interesting and added a paragraph to Section 5.1 and a new Figure with the singular value plot with and without the erroneous sensor. This clearly shows that the issue is resolved by removing that sensor.

➔ In contrast to Musselman Arch, the higher singular values at Corona Arch (but also at Rainbow Bridge and Squint Arch) are peaking above the noise floor, indicating the presence of a real mode.

*5) Finally, I apologize to insist on this point: I had asked further details of the "active" experiment. The authors referred to "Geimer et al. (2020) (who) applied this technique to a set of small-sized natural rock arches by stomping on the ground next to the structure and applying a band-pass filter around the resonant frequency". I did not find more details on the paper of Geimer et al (2020) (acknowledge the authors that facilitated it). Then, the authors may find here the place to develop.*

*First, eq (3) is not derived from eq (2), unless "small" damping (how much small ? 1%, 10%...). Second, y(t) is the amplitude of what exactly ? From the text (Line 192), it seems simply the recordings (just after the stomp) filtered around the resonant frequency of the mode. Several questions then arise : which sensor(s) ? All of them ? Average ? How many time windows used ? How far away from the arch ? What about source deconvolution (as from Figure 2b it is written IRF) ? Can the stomp be enough to excite fundamental modes at rather low frequencies (< 15 Hz) ?... It would be much simpler to explicitly show the example of Squint arch (logarithmic decay of active "stomp" data). The authors state that the active experiments can be considered "good estimates of damping" (Lines 190-195). This must still be demonstrated.*

➔ Equation 3 is an approximation of Equation 2 for small damping ratios. We replaced the equal sign in Equation 3 by an approximation sign.

➔ There is no limit of what is small, as this depends on the uncertainty tolerance of the analysis. When evaluating Equations 2 and 3 it is evident that the error is smaller than 2% for damping values smaller than 20%, which covers most practical structures (Chopra, 2015).

➔ For damping ratios smaller than 3%, as it is observed on all rock arches in this study, the error is as small as 0.04%.

➔ $y(t)$ is the amplitude of the measured quantity as a function of time and can be deformation, velocity or acceleration. As defined in Section 2, we are recording velocities. Specification added.

- ➔ It is not the scope of this paper to demonstrate that performing active experiments is a good technique for measuring damping of a structure. Measuring the decaying amplitude of a vibrating structure is the most direct measurement of damping (the decay of the amplitude is the damping). The problem with active experiment is that it is not always possible (or allowed) to excite the structure artificially, which is the reason why ambient vibration techniques are used (e.g., Magalhães et al., 2010). We added a sentence to Section 3.1.
- ➔ This study is about the demonstration that the EFDD and SSI-COV modal analysis techniques can be used to determine the modal properties of rock arches. This study is not about determining damping by active experiments and we do not want to overload our publication by discussing techniques that are not required for our study.
- ➔ Geimer et al. (2020) used the active experiment to verify their results obtained by the half-power bandwidth. Therefore, in the revised manuscript, we avoid direct comparison to the active experiment and only mention the comparison to the half-power bandwidth method (which is the same value). This shows that our study is not relying on this active experiment and that our manuscript would contain the same results and conclusions if that active experiment was never performed.
- ➔ As a reply to the reviewer, we show the raw (i.e. unfiltered) data of one stomp below. The dominant frequency is here simply determined by the inverse of the median of the natural periods determined by picking the maxima on the trace (red circles). The dominant frequency is ~12.5 Hz, corresponding to f2 in our study. Note that Geimer et al. (2020) interpreted f1 and f2 as one single fundamental mode.

[Figure]

[Figure]

[Figure]

**REFERENCES**

Chopra, A. K. (2015). *Dynamics of structures : theory and applications to earthquake engineering* (fourth ed.): Boston : Pearson Prentice Hall.

Geimer, P. R., Finnegan, R., & Moore, J. R. (2020). Sparse Ambient Resonance Measurements Reveal Dynamic Properties of Freestanding Rock Arches. *Geophysical Research Letters, 47*(9), e2020GL087239. https://doi.org/10.1029/2020GL087239

Magalhães, F., Cunha, Á., Caetano, E., & Brincker, R. (2010). Damping estimation using free decays and ambient vibration tests. *Mechanical Systems and Signal Processing, 24*(5), 1274-1290. https://www.sciencedirect.com/science/article/pii/S0888327009000727